

# ORCHIDEE-SOM: Modeling soil organic carbon (SOC) and dissolved organic carbon (DOC) dynamics along vertical soil profiles in Europe

Marta Camino-Serrano[1,2], Bertrand Guenet[3], Sebastiaan Luyssaert[4], Philippe Ciais[3], Vladislav Bastrikov[3], Bruno De Vos[5], Bert Gielen[6], Gerd Gleixner[7], Albert Jornet-Puig[3], Klaus Kaiser[8], Dolly Kothawala[9], Ronny Lauerwald[10],

Josep Peñuelas[1,2], Marion Schrumpf[7], Sara Vicca[6], Nicolas Vuichard[3], David Walmsley[11], Ivan A. Janssens[6]

[1] CREAF, Cerdanyola del Vallès, 08193, Catalonia, Spain

[2] CSIC, Global Ecology Unit CREAF-CSIC-UAB, Bellaterra 08193, Catalonia, Spain

[3] Laboratoire des Sciences du Climat et de l'Environnement, LSCE/IPSL, CEA-CNRS-UVSQ, Université Paris-Saclay, F-91191 Gif-sur-Yvette, France

[4] Department of Ecological Sciences, Free University Amsterdam (VUAmsterdam), De Boelelaan 1085, 1081 HV Amsterdam, the Netherlands

[5] INBO, Research Institute for Nature and Forest, Gaverstraat 4, 9500 Geraardsbergen, Belgium

[6] Department of Biology, Research Group of Plant and Vegetation Ecology, University of Antwerp, Universiteitsplein 1, B-2610 Wilrijk, Belgium

[7] Max-Planck-Institute for Biogeochemistry, Hans-Knöll-Straße 10, 07745 Jena, Germany

[8] Soil Science and Soil Protection, Martin Luther University Halle-Wittenberg, von-Seckendorff-Platz 3, 06120 Halle (Saale), Germany

[9] Department of Limnology, Evolutionary Biology Centre, Norbyvägen 18D, Uppsala University, Uppsala, SE-75236, Sweden

[10] Department of Mathematics, College of Engineering, Mathematics and Physical Sciences, University of Exeter, Exeter EX4 4QE, UK

[11] Institute of Ecology, Faculty of Sustainability, Leuphana University of Lüneburg, Universitätsallee 1, D-21335, Germany

*Correspondence to*: Marta Camino-Serrano (m.camino@creaf.uab.cat)

**Abstract.** Current Land Surface Models (LSMs) typically represent soils in a very simplistic way, assuming soil organic carbon (SOC) as a bulk, thus impeding a correct representation of deep soil carbon dynamics. Moreover, LSMs generally neglect the production and export of dissolved organic carbon (DOC) from soils to rivers, leading to overestimations of the potential carbon sequestration on land. These common oversimplified processing of SOC

in LSMs is partly responsible for the large uncertainty in the predictions of the soil carbon response to climate change. In this study, we present a new soil carbon module called ORCHIDEE-SOM, embedded within the land surface model ORCHIDEE, which is able to reproduce the DOC and SOC dynamics in a vertically discretized soil to two meters. The model includes processes of biological production and consumption of SOC and DOC, DOC adsorption on- and desorption from soil minerals, diffusion of SOC and DOC and DOC transport with water

through and out of the soils to rivers. We evaluated ORCHIDEE-SOM against observations of DOC concentrations and SOC stocks from four European sites with different vegetation covers: a coniferous forest, a deciduous forest, a grassland and a cropland. The model was able to reproduce the SOC stocks along their vertical profiles at the four sites and the DOC concentrations within the range of measurements, with the exception of the DOC



concentrations in the upper soil horizon at the coniferous forest. However, the model was not able to fully capture the temporal dynamics of DOC concentrations. Further model improvements should focus on a plant- and depth-dependent parameterization of the new input model parameters, such as the decomposition times of DOC and the microbial carbon use efficiency. We suggest that this new soil module, when parameterized for global simulations,

5    will improve the representation of the global carbon cycle in LSMs, thus helping to constrain the predictions of the future SOC response to global warming.





## 1 Introduction

The soil is the largest terrestrial carbon pool and its response to global warming is thus crucial for the global carbon (C) cycle and its feedback to climate change (Jobbagy and Jackson, 2000; Todd-Brown et al., 2014). Among other things, Earth System Models aim to predict the vulnerability of the global carbon cycle to climate change. However

to date the response of the soil organic carbon (SOC) pool and fluxes to climate change remains highly uncertain, mainly because the mechanistic understanding of the soil processes remains imperfect, and because new knowledge is not rapidly incorporated in these global models (Bradford et al., 2016; Schmidt et al., 2011; Todd-Brown et al., 2013; Wieder et al., 2015). In this sense, many authors claimed the need for a more realistic representation of what governs SOC dynamics and transport within land surface models (LSMs), which are the

land component of the Earth system models (Battin et al., 2009; Nishina et al., 2014; Schmidt et al., 2011; Todd-Brown et al., 2014; Wieder et al., 2015). Among the suggested model improvements, the representation of the vertical SOC distribution and dissolved organic carbon (DOC) in the soil and the lateral export of carbon out of the soil are most likely to considerably improve model simulations.

Deep (> 1 m) soil C accounts for more than half of the global SOC stocks and it is stabilized for long periods (from

decades to millennia) (Jobbagy and Jackson, 2000; Koarashi et al., 2012). Recent studies have, however, shown that environmental changes may destabilize deep SOC, for instance, by accelerating decomposition when labile organic carbon is provided to the microbial community (Fontaine et al., 2007), making it highly vulnerable to the primary production increase associated to global change or to the modification of root profiles due to land use change (Hurtt et al., 2011; Norby et al., 2005; Ryan et al., 2017). Two main processes cause the vertical movement

of SOC into deep soils: dispersal of SOC during mixing, which is represented in models as a diffusion process, and is mainly due to bioturbation caused by animal and plant activity in soil and to cryoturbation in permafrost soils; and advection, which is the transport of carbon with the liquid phase moving through the soil, affecting only the soluble C pool (Braakhekke et al., 2013). Nevertheless, the flux of C into deep layers is difficult to model because multiple mechanisms co-occur, which hampers the isolation of their effects from SOC profile

measurements only (Braakhekke et al., 2013).

DOC is one of the main sources of subsoil SOC and, at the same time, an important substrate for microorganisms in deep soils, particularly under humid conditions (Kaiser and Kalbitz, 2012; Neff and Asner, 2001; Rumpel and Kogel-Knabner, 2011). DOC may be strongly retained in mineral soils by adsorption and may thus contribute to SOC sequestration (Schrumpf et al., 2013). On the other hand, because soil microbial activity in deep layers is

limited by fresh and labile substrates, the input of fresh DOC may stimulate SOC decomposition in deep soils (Fontaine et al., 2007; Rumpel and Kogel-Knabner, 2011). Therefore, besides transport in the soil column by advection and diffusion, two main processes control DOC dynamics: 1) Biological production and consumption and 2) Adsorption to and desorption from soil minerals, and these processes in turn impact SOC cycling along the soil profile (Dwivedi et al., 2017; Kaiser and Kalbitz, 2012).

Despite the importance of deep soil carbon, to date only one LSM (CLM4) incorporates mechanisms for vertical mixing and subsequent stabilization of carbon (Koven et al., 2013). Representation of SOC in LSMs is generally highly simplified, with a single-layer box modelling approach and without representation of a soluble pool of carbon (Todd-Brown et al., 2013). This approach assumes that deeper SOC and DOC do not play an active role in the C cycle and thus hampers the prediction of the soil feedback to global warming. Hence, it seems clear that

more realistic representations of the mechanisms controlling SOC stocks and DOC processing and transport along



the soil profile are necessary to predict the vulnerability of deep SOC to climate change in order to accurately model the future soil carbon stock trajectories.

Furthermore, the DOC that is not retained in the soils is displaced along the land-aquatic continuum (Battin et al., 2009; Le Quéré et al., 2014) and it is a main substrate for freshwater decomposition. Modelling the DOC exported from soils to aquatic systems is thus important for accurate estimations of C budgets, as it corresponds to a fraction of the carbon taken up from the atmosphere that is not sequestered in soils. Moreover, this fraction is altered due to anthropogenic activity (Le Quéré et al., 2014; Regnier et al., 2013). Although the magnitude of C losses through DOC export is very small compared to the gross ecosystem carbon fluxes (between 2% and 5% of soil heterotrophic respiration after Regnier et al., 2013; Schulze et al., 2009), neglecting the DOC export from land in LSMs can lead to a systematic overestimation of the SOC stocks and of the SOC sink (Jackson et al., 2002; Janssens et al., 2003).

Several models can predict DOC concentrations and export at site, landscape or catchment scale (Ahrens et al., 2015; Futter et al., 2007; Gjettermann et al., 2008; Jason C. Neff and Gregory P. Asner, 2001; Jutras et al., 2011; Michalzik et al., 2003; Ota et al., 2013; Tian et al., 2015; Wu et al., 2013). These models differ in the definitions of the soil C pools (from turnover times to chemically differentiated fractions), the level of detail in the process formulation (e.g., from simple first order kinetics to non-linear relationships, including or not including sorption to soil minerals) and the spatial and temporal resolution (from site to global, and from hourly to annual or longer time scales). While these models have been successfully tested, and are able to reasonably simulate DOC dynamics, at present, only two models exist that can predict DOC export from soil at global scale (Langerwisch et al., 2016; McGuire et al., 2010) and there is no global LSM embedded within an Earth System Model that represents a vertically resolved module of SOC and DOC production, consumption, sorption and transport.

The purpose of this paper is to describe the new soil C module ORCHIDEE-SOM, embedded in the LSM ORCHIDEE, which is able to reproduce the SOC and DOC dynamics in a vertically discretized soil down to two meters, consistent with water transport and soil thermics. We also perform a first evaluation of the new soil module ability to reproduce SOC stocks and DOC concentrations dynamics by comparing model predictions with respective field observations at four European experimental sites with different vegetation covers and soil properties. If the model structure is valid, ORCHIDEE-SOM should be able to reproduce, not only the values of DOC and SOC concentrations within the range of the observations, but also the internal soil processes that drive the site-specific differences in SOC stocks following differences in soil texture, vegetation and climate and the decrease in SOC and DOC down the soil profile.

## 2 Model developments

ORCHIDEE-SOM is an extension to the soil module in ORCHIDEE, based on the ORCHIDEE version SVN r3340. ORCHIDEE represents the principal processes influencing the carbon cycle (photosynthesis, ecosystem respiration, soil carbon dynamics, fire, etc.) and energy exchanges in the biosphere (Krinner et al., 2005). It consists of two modules: SECHIBA, describing the fast processes of energy and water exchanges between the atmosphere and the biosphere at a time step of 30 minutes (de Rosnay et al., 2002) and STOMATE, which calculates the phenology and carbon dynamics of the terrestrial biosphere at a time step of one day. ORCHIDEE represents vegetation globally using 13 Plant Functional Types (PFT): one PFT for bare soil, eight for forests, two for grasslands, and two for croplands (Krinner et al., 2005).



In the trunk version of ORCHIDEE, soil carbon is based on the CENTURY model following Parton et al., (1988). Accordingly, soil carbon is divided in two litter pools (metabolic and structural) and three soil organic carbon (SOC) pools (slow, active and passive) based on SOC stability, each with different turnover rates. The decomposition rate of each pool is controlled by temperature, moisture and clay content, and results in carbon

fluxes from the litter to the SOC as well as from and between the three SOC pools. The fraction of the decomposed carbon being transferred from one pool to another is prescribed using parameters based on Parton et al., (1988) (Table 1) and the rest is lost to the atmosphere as heterotrophic respiration. The vertical distribution of SOC with particular dynamics at each depth is not considered and losses of soil carbon by dissolution and transport are not represented in the model (Fig. 1).

ORCHIDEE-SOM upgrades the trunk version of ORCHIDEE to simulate carbon dynamics in the soil column down to two meters depth, partitioned in 11 layers following the same scheme as in the hydrological module ORC11 (Campoy et al., 2013; Guimberteau et al., 2014). ORCHIDEE-SOM mechanistically models the concentration of DOC in each soil layer and its transport between layers. Moreover, the upgraded module links SOC decomposition with the amount of fresh organic matter as a way of accounting for the priming effect (Guenet

et al., 2017).

In short, ORCHIDEE-SOM represents four litter pools (metabolic aboveground litter, metabolic belowground litter, structural aboveground litter and structural belowground litter) and three SOC pools based on their turnover rate (active, slow and passive). Two new pools were added to represent the DOC defined by their decomposition rate: the labile and the stable DOC, with a high and low decomposition rate, respectively. Each pool may be in the

soil solution or adsorbed on the mineral matrix. The products of litter- and SOC decomposition go to free DOC, which, in turn, is decomposed following first order kinetics equation (e.g., Kalbitz et al., 2003; Qualls and Haines, 1992b). One part of the decomposed DOC goes back to SOC pools, according to a fixed microbial carbon use efficiency ($CUE_{DOC}$) parameter, the other part is converted into $CO_2$ and contributes to heterotrophic respiration. The free DOC can then be adsorbed to soil minerals or remain in solution following an equilibrium distribution

coefficient ($K_D$) (Nodvin et al., 1986), which depends on soil properties (clay and pH). Adsorbed DOC is assumed to be protected and thus it is neither decomposed nor transported within the soil column. Free DOC is subject to transport with the water flux between layers calculated by the soil hydrological module of ORCHIDEE, i.e. by advection. Also, SOC and DOC are subject to diffusion, that is represented using the second Fick's law of diffusion. All the described processes occur within each soil layer. At the end of every time step, the flux of DOC

(expressed in g C m$^{-2}$ d$^{-1}$) leaving the soil with runoff (upper layer) and drainage (bottom layer) is calculated by multiplying DOC concentrations in the solution with the runoff and drainage flux calculated by the hydrological module ORC11 (Fig. 1).

This section presents the new ORCHIDEE-SOM formulations in more detail, focusing first on the vertical discretization scheme, followed by the newly implemented biological and physical processes affecting soil carbon

(i.e., decomposition, sorption of DOC, advection and diffusion) and, finally, the model parameterization and evaluation exercise are described.

## 2.1. Vertical discretization of the soil carbon module

For mathematical reasons, ORCHIDEE SVN r3340 has two different discretized schemes for soil physics: one for energy and other for hydrology. Since ORCHIDEE-SOM requires the transport of water between layers and





drainage for the calculation of DOC concentrations and fluxes, we adopted the discretization used for the soil hydrology scheme whose performance has already been tested against tropical (Guimberteau et al., 2014), boreal (Gouttevin et al., 2012) and temperate datasets (Campoy et al., 2013). Therefore, ORCHIDEE-SOM represents a two-meter soil column with 11 discrete layers of geometrically increasing thicknesses with depth. This kind of

geometric configuration is used in most LSMs describing the vertical soil water fluxes based on the Richards equation, such as ORCHIDEE (Campoy et al., 2013). More information on the hydrological formulation of ORCHIDEE is given in section 2.2.3.

The midpoint depths (in meters from the surface) of the layers in the discretized soil column are: 0.00098, 0.00391, 0.00978, 0.02151, 0.04497, 0.09189, 0.18573, 0.37341, 0.74878, 1.49951, respectively. The first layers in the soil

hydrology discretization scheme are thinner (1 mm) than needed in terms of biological and pedogenetic process representation. Nevertheless, we decided to integrate the 11-layers scheme for technical reasons: it simplifies the coding and the understanding of the code for the users. At every time step, each soil layer is updated with all the sources and sinks of DOC due to the represented biological and physical processes.

The new 11-layers scheme applies to the soil carbon in the mineral soil and for the belowground litter (see Section

2.2.1.). However, the aboveground litter layer in ORCHIDEE is dimensionless, which means that processes of production and decomposition of aboveground litter occur independently of the litter layer thickness in the model. In ORCHIDEE-SOM, a new parameter to define the thickness of the aboveground litter layer ($z\_litter$), assumed constant over time, has been added to allow the calculation of aboveground litter diffusion into the mineral soil (Table 1).

**2.2. Biological and physical processes affecting SOC and DOC**

**2.2.1. Litter, SOC and DOC dynamics within each soil layer**

In ORCHIDEE-SOM, litter is defined by two pools called metabolic and structural with high and low decomposition rates, respectively. Above- and belowground litter are separate pools. While belowground litter is discretized over the 11-layers scheme down to two meters, aboveground litter layer is simply defined by a fixed

thickness parameter (Table 1). The litter is distributed belowground following an exponential root profile with different root density profile parameter ($\alpha$) for each PFT.

$$rp = 1/(1 - e^{(-depth/\alpha)}) \tag{1}$$

with $rp$ being the root profile, *depth* the maximum depth of the model (fixed to two meters) and $\alpha$ a PFT parameter dependent (in meters). Litter (*LitterC*) decomposition for each pool $i$ (aboveground metabolic, aboveground structural, belowground metabolic and belowground structural) is described by first order kinetics (Eq. 2):

$$\frac{\partial LitterC_{i,z}}{\partial t} = I(t)_{i,z} - k_{LitterCi} \times LitterC_{i,z}(t) \times \theta(t) \times \tau(t) \tag{2}$$

with $I$ being the carbon input coming from deceased plant tissues in g C m$^{-2}$ ground days$^{-1}$ and $k_{LitterC}$ the litter decomposition rates in days$^{-1}$, which are fixed and similar to the rates used for SOC in ORCHIDEE SVN r3340





(Table 1). The litter decomposition is affected by two rates modifiers, $\theta$ and $\tau$, to take into account the effect of moisture and temperature, respectively:

$$\theta = \max\left(0.25, \min(1, 1.1 \times M^2 + 2.4 \times M + 0.29)\right) \qquad (3)$$

$$\tau = \min\left(1, e^{0.69 \times (T-303.15)/10}\right) \qquad (4)$$

With $M$ and $T$ being the soil moisture (m³ m⁻³) and the temperature (K) of the layer considered. For the aboveground litter (dimensionless), averaged moisture and temperature over the four first layers are used to calculate the rate modifiers given by Eq. 3 and 4.

The SOC is defined by three pools, so called active, slow and passive, with different turnover rates. The SOC decomposition is based on Guenet et al., (2013):

$$\frac{\partial soc_{i,z}}{\partial t} = I(t)_{i,z} - k_{SOC,i} \times (1 - e^{-c \times LOC(t)}) \times SOC(t)_{i,z} \times \theta(t) \times \tau(t) \qquad (5)$$

with $I$ being the carbon input into the pool $i$ considered for each soil layer $z$ coming from litter and DOC in g C m⁻² days⁻¹, $k_{SOC}$ a SOC decomposition rate (days⁻¹), $LOC$ the stock of labile organic C defined as the sum of the C pools with a higher decomposition rate than the pool considered. This means that for the active carbon pool $LOC$ is the litter and DOC, but for the slow carbon pool $LOC$ is the sum of the litter, DOC and active SOC pools, and finally, for the passive carbon pool $LOC$ is the sum of litter, DOC, active and slow SOC pools. Finally, $c$ is a parameter controlling the impact of the $LOC$ pool on the SOC mineralization rate, i.e., the priming effect (Guenet et al., 2016, Table 1). The decomposition of the active SOC pool is further modified by a clay modifier $\gamma$, which considers that the SOC decomposition decreases when increasing the clay content:

$$\gamma = 1 - 0.75 \, x \, clay \qquad (6)$$

In reality, DOC is produced from soil microbial biomass, litter, soil organic carbon, root exudates and desorption from minerals. In ORCHIDEE-SOM, all the products of decomposition from litter and SOC go to free DOC. We assumed that root exudates are represented within the decomposed belowground metabolic litter and that the root-derived material comes within the products of decomposition of the belowground structural litter. In ORCHIDEE-SOM, like in most of the current global-scale LSMs, soil microbial biomass is not explicitly represented (Schmidt et al., 2011; Todd-Brown et al., 2013). Instead, we assumed that all DOC is taken up by microorganisms, which then use a certain portion of organic C for respiration, and the rest for growth (controlled by the microbial carbon use efficiency), which eventually ends up as dead microbial biomass in the SOC pools (Gleixner, 2013).

In the model, DOC is represented using two pools that are defined by their decomposition rates; the labile DOC pool with a high decomposition rate, and the stable DOC pool with a lower decomposition rate (Table 1, Turgeon, (2008)). The labile pool corresponds to the DOC coming from litter and active carbon, while the stable pool corresponds to the DOC coming from slow and passive carbon. The DOC pools in the model can be *free* in the soil solution or *adsorbed* to the soil minerals. To avoid extremely high and unrealistic DOC concentrations in the





first very thin soil layer, the DOC coming from aboveground litter decomposition is re-distributed among the first five soil layers, which represent the first 5 cm of soil approximately. Only the free DOC is decomposed in the model, following first order kinetics equation (Eq. 7), a classical approach to describe DOC decomposition (Kalbitz et al., 2003). Therefore, the change in DOC for each pool *i* due to biological activity at each layer and

every time step is described as:

$$\frac{\partial DOC_{i,z}}{\partial t} = I(t)_{i,z} - k_{DOC,i} \times CUE_{DOC} DOC(t)_{i,z} \tag{7}$$

With *I* being the input coming from litter and SOC decomposition (described above) in g C m$^{-2}$ ground days$^{-1}$, $k_{DOC}$

a parameter representing the decomposition rate of free DOC pool *i* (labile, and stable) in days$^{-1}$, which corresponds to the inverse of the *DOC_tau_labile* or *DOC_tau_stable* parameters in ORCHIDEE-SOM (Table 1). The decomposed DOC (second term of Eq. 7) is partially respired and partially redistributed in the SOC pools, the fraction of respired DOC (Resp$_{DOC}$) in g C m$^{-2}$ day$^{-1}$ being controlled by the carbon use efficiency (CUE$_{DOC}$) parameter, which remains constant for all paths from DOC to SOC pools (Table 1, Eq. 8):

$$Resp_{DOC,i,z}(t) = (1 - CUE_{DOC})x\ k_{DOC,i} \times DOC(t)_{i,z} \tag{8}$$

The not-respired, *recycled* DOC (DOC$_{Recycled}$) coming from active, slow and passive SOC pools are redistributed in the different SOC pools following the same parameters as in the CENTURY model (Guenet et al., 2016; Parton

et al., 1988):

$$DOC_{Recycled,i,j}(t) = fra\_carb\_ij \times CUE_{DOC} \times K_{DOC,i} \times DOC(t)_{i,z} \tag{9}$$

With DOC$_{Recycled,i,j}$ being the DOC flux going back from pool *i* to pool *j* and *frac_carb_ij* the prescribed fraction

of carbon from pool *i* to *j* (Table 1).

### 2.2.2. DOC sorption to soil minerals

DOC retention in mineral soils is largely driven by abiotic processes of adsorption and desorption. Most of the DOC models commonly represent adsorption using the simple Initial Mass (IM) linear isotherm (Eq. 10, Neff and Asner, 2001; Wu et al., 2013) or using a first order kinetic reaction to represent a linear adsorption (Laine-Kaulio

et al., 2014; Michalzik et al., 2003):

$$DOC_{RE} = m \times DOC_i - b \tag{10}$$

With *DOC$_{RE}$* being the amount of DOC desorbed (negative value) or adsorbed (positive value), *m* a regression

coefficient similar to the partitioning coefficient, *DOC$_i$* the initial concentration of free DOC in solution and *b* the intercept (the desorption parameter) in g kg$^{-1}$ soil.

In principle, the IM and linear approaches are expressions of a simple partitioning process, where the tendency of the soil to adsorb DOC is described by an equilibrium partition coefficient (K$_D$). Hence, K$_D$ is defined as a measure of the affinity of the substances for the soil when the reactive substance present in the soil (DOC in our case) is





assumed to be insignificant. The equilibrium partition coefficient can be related to the regression coefficient $m$ in the IM isotherm following (Nodvin et al., 1986) by Eq. (11):

$$K_D = \frac{m}{1-m} \times \frac{(volume\ of\ solution)}{(mass\ of\ soil)} \qquad (11)$$

Where $K_D$ ($m^3$ $kg^{-1}$ soil) represents the equilibrium distribution between the adsorbed and free dissolved organic carbon, and thus, will vary depending on the adsorption capacity of the soil profile.

ORCHIDEE-SOM assumes that adsorption/desorption occurs due to the deviation between the actual concentration of adsorbed DOC and the equilibrium adsorbed DOC defined by $K_D$. Therefore, the DOC adsorption
in soil minerals in ORCHIDEE-SOM is formulated as follows:

$$DOC_{RE-EQ} = K_D \times DOC_i(t) \times BD \times \frac{1}{\theta} \qquad (12)$$

$$\frac{\partial DOC_i}{\partial t} = DOC_i(t) - (DOC_{RE-EQ}(t) - DOCad_i(t)) \qquad (13)$$

$$\frac{\partial DOCad_i}{\partial t} = DOCad_i(t) + (DOC_{RE-EQ}(t) - DOCad_i(t)) \qquad (14)$$

In Eq. 12, $DOC_{RE-EQ}$ is the amount of adsorbed DOC in equilibrium according to the partition coefficient $K_D$. As $K_D$ is expressed in $m^3$ $kg^{-1}$ soil, we use bulk density (BD) and soil moisture ($\theta$) to convert DOC from g C $kg^{-1}$ soil into g C $m^{-3}$ water. $DOC_i(t)$ and $DOCad_i(t)$ are the concentration of total DOC and the concentration of adsorbed DOC for each pool (labile and stable) in g C $m^{-3}$ water, respectively.
This approach assumes that the free DOC produced at every time step of the model (30 minutes) is immediately distributed between the adsorbed and free pools to reach equilibrium, in agreement with studies showing that sorption occurs rapidly, within seconds to minutes (Kothawala et al., 2008; Qualls and Haines, 1992a).

***Dependence of the sorption distribution coefficient on soil properties***
The adsorption characteristics of soils have previously been related to several soil properties. For instance, the desorption parameter ($b$) of the IM isotherm (Eq. 10) has been related to the organic carbon content in the soil profile whereas the partition coefficient ($m$) was related to oxalate-extractable aluminium ($Al_o$) and dithionite-extractable iron ($Fe_d$) and organic carbon content (Kaiser et al., 1996). Also, the maximum adsorption capacity of a soil was found to correlate to Fe and Al in soil (Kothawala et al., 2008). Despite the accepted importance of Al
and Fe in controlling DOC dynamics in soils (Camino-Serrano et al., 2014), these variables are not globally available and hence not included as boundary conditions for the land surface model ORCHIDEE. Therefore, in order to produce a statistical model that predicts the $K_D$ parameter as a soil-condition dependent variable, we focused on other soil parameters available within ORCHIDEE that correlated with the $K_D$ coefficient of DOC sorption in soils and are indicative of Al and Fe in the soil, which are clay, organic carbon (OC), and pH (e.g.,
Jardine et al., 1989; Kaiser et al., 1996; Moore et al., 1992).

We calculated the distribution coefficient $K_D$ from the IM isotherm partition coefficient ($m$) measured in batch experiments on 34 European soil profiles (Kaiser et al., 1996), according to Eq. 11, and built an empirical model that related $K_D$ with soil depth, clay, and pH. We selected the best model by means of stepwise regressions. The distribution of the residuals was tested and models whose residuals were not normally distributed were discarded.





The selected model included only clay and soil pH as explanatory variables and explained 25% of the variability in $K_D$ (adjusted $R^2 = 0.25$, Fig. S1) (Eq. 15):

$$\log K_D = 0.001226 - 0.000212 * pH + 0.00374 * Clay \tag{15}$$

By using this relationship, the effects of soil texture and pH on the adsorption capacity of the soil are represented empirically in the model.

### 2.2.3. Vertical fluxes of SOC and DOC

ORCHIDEE-SOM assumes that SOC and DOC move along the soil profile as a result of three processes:
bioturbation results in vertical fluxes of SOC, and diffusion and advection produces vertical fluxes of DOC.

#### *Diffusion*

In general, bioturbation, which is defined as the transport of plant debris and soil organic matter by soil fauna, causes homogenization of soil properties, i.e., net transport of soil constituents proportional to the concentration
gradient. Therefore, the effects of bioturbation on the distribution of soil properties is commonly represented in models as a diffusion process using Fick's diffusion equation (e.g., Braakhekke et al., 2011; Elzein and Balesdent, 1995; Obrien and Stout, 1978; Wynn et al., 2005). However, some conditions must be respected to use Fick's law to bioturbation. 1) the time between mixing events must be short compared to other processes. 2) the size of each layer must be small compared to the total depth of the profile and 3) the mixing should be isotropic (bottom-up
and top-down) (Braakhekke et al., 2011). If these conditions are fulfilled, bioturbation can lead to diffusive behaviour of soil constituents and can be represented by Fick's diffusion law (Boudreau, 1986). At small spatial scales, bioturbation may not meet these criteria, but at sufficiently large spatial scales, the assumption of diffusive behaviour is reasonable (Braakhekke et al., 2011). Hence, we assume that bioturbation can be modelled as a diffusion process at global scale, for which ORCHIDEE-SOM is designed.
Therefore, in ORCHIDEE-SOM, we represented bioturbation by a diffusion equation based on Fick's second law (Eq. 16):

$$F_D = -D \times \frac{\partial^2 c}{\partial z^2} \tag{16}$$

where $F_D$ is the flux of C transported by diffusion in g C $m^{-3}$ soil $day^{-1}$, $-D$ the diffusion coefficient ($m^2$ $day^{-1}$) and $C$ the amount of carbon in the pool subject to transport (g C $m^{-3}$ soil). In ORCHIDEE-SOM, bioturbation represented as diffusion applies to the SOC pools and the belowground litter and the diffusion coefficient is assumed to be constant across the soil profile in ORCHIDEE-SOM (Table 1).
DOC may also be transported by diffusion following Eq. 16, but with different diffusion coefficient ($D\_DOC$)
(Table 1). Unlike for SOC and belowground litter, diffusion of DOC is not due to bioturbation processes, but a representation of DOC movement due to actual diffusion, that is, movements of molecules due to concentration gradients. For this, we assume that the water distribution is continuous along the soil column (i.e., there are no dry places), assumption that does not always hold true nature, where actually DOC is transported through preferential flow pathways, determined by worm holes and other biogalleries.





*Advection*

Like most models, ORCHIDEE-SOM represents the transport of carbon with the liquid phase (only DOC in our case) by means of advection (e.g., Braakhekke et al., 2011; Futter et al., 2007). The calculation of advection fluxes in ORCHIDEE-SOM relies on the flux of water between soil layers as calculated by the soil hydrology module ORC11, which is briefly described hereafter.

The soil hydrology module is based on the two-meters vertical discretization of the soil column (see section 2.1.). A physically-based description of the unsaturated water flow was introduced in ORCHIDEE by de Rosnay et al., (2002). Soil water fluxes calculation relies on a one-dimensional Fokker-Planck equation, combining the mass and momentum conservation equations using volumetric water content as a state variable (Campoy et al., 2013). Due to the large scale at which ORCHIDEE is applied, the lateral fluxes between adjacent grid cells are neglected. Also, all variables are assumed to be horizontally homogeneous. The flux field $q$ along the soil profile comes from the equation of motion known as Darcy (1856) equation in the saturated zone, and extended to unsaturated conditions by Buckingham, (1907):

$$q(z,t) = -D\big(\theta(z,t)\big)\frac{\partial \theta(z,t)}{\partial z} + K(\theta(z,t)) \tag{17}$$

In this equation, z is the depth (in m) below the soil surface, t (in s) is the time, $K(\theta)$ (in m s$^{-1}$) is the hydraulic conductivity and $D(\theta)$ (in m$^2$ s$^{-1}$) is the diffusivity.

The soil hydrological module counts with the following boundary conditions at the soil surface and at the bottom layer. First, the water flux at the soil surface is defined by the difference between infiltration into the soil and soil evaporation. Precipitation is partitioned between surface runoff and infiltration into the soil, by characterizing the wetting front speed through the top soil layers (d'Orgeval et al., 2008). Soil evaporation is calculated assuming that it can proceed at the potential rate, unless water becomes limiting. Second, ORCHIDEE assumes conditions of free gravitational drainage at the soil bottom. This boundary condition implies that soil moisture is constant below the lower node, which is not always the case in nature. In particular, when a shallow water table is present, water saturation within the soil column cannot be modelled within ORCHIDEE. More information on the calculation of the water flux, runoff and drainage can be found in Campoy et al., (2013).

The transport of DOC within the liquid phase is assumed to occur due to advection flux and it is modelled as the flow of water calculated by the hydrology module multiplied by the concentration of DOC at each layer according to Eq. 18 (Futter et al., 2007) :

$$F_A = A \times DOC_i \tag{18}$$

With $F_A$ the advection flux of free DOC in g C m$^{-2}$ 30 min$^{-1}$, A the flux of water calculated by the hydrological module in kg m$^{-2}$ 30 min$^{-1}$, and $DOC_i$ the concentration of DOC free in solution in pool $i$ in g C m$^{-3}$ water.

At every time step, DOC in each layer is updated with the DOC fluxes entering and leaving the soil layer. The final DOC concentration in the last and the first five layers is multiplied by drainage and runoff, respectively, to calculate the amount of DOC leaving the system (in g C m$^{-2}$ ground).





### 2.3. Model parameterization and evaluation

#### 2.3.1. Sites description

Four European sites with available data on soil DOC concentrations and SOC stocks were selected for the validation of ORCHIDEE-SOM. The four sites correspond to four different PFTs: Brasschaat, a coniferous forest,

Hainich, a deciduous forest, and two experimental sites in Carlow: the so called "Lawn Field", a grassland and the "Pump Field", a cropland (hereafter referred to as Carlow grassland and Carlow cropland, respectively). Brasschaat forest (Belgium, 51°18′N, 4°31′E) is a Scots pine (*Pinus sylvestris*) forest growing in a sandy soil classified as Albic Hypoluvic Arenosol (Gielen et al., 2011; Janssens et al., 1999). Hainich forest (Germany, 51°4'N,10°27'E) is a forest with beech (*Fagus sylvatica*) as the dominant species growing in a clayey soil

classified as Eutric Cambisol (Kutsch et al., 2010; Schrumpf et al., 2013). In Carlow grassland (Ireland, 52°52'N, 6 º54'W) a mixture of ~70% perennial ryegrass and 30% white clover was sown in a loamy soil classified as Calcic Luvisol (Walmsley, 2009). Finally, Carlow cropland (Ireland, 52°51'N, 6 º55'W) is an arable site that has been under crop rotation for the last 45 years, being under spring barley cultivation from 2000 to the present. The soil in Carlow cropland is classified as Eutric Cambisol (Schrumpf et al., 2013; Walmsley et al., 2011). Therefore,

these four sites cover a wide range of vegetation cover and soil properties (from acidic to circumneutral soils, and from sandy to clayey soils) (Table 2).

At the four sites, measurements of DOC concentrations were available for at least one year (Table 2). DOC concentrations were typically measured fortnightly, except for periods when sites could not be reached or when soils were too dry to extract water. DOC concentrations were available at more than one soil horizon for all sites,

except for Carlow cropland. In Carlow grassland, there were two sampling positions (Box 1 and Box 2), located next to each other but substantially differing in some soil properties, like texture and soil water content (Walmsley, 2009). More detailed information about the sampling and analysis for DOC concentrations can be found in Gielen et al., (2011) and Kindler et al., (2011).

SOC concentrations along the soil profile were measured at Brasschaat, Hainich and Carlow cropland by dry

combustion (TC analyser) (Schrumpf et al., 2013). At Hainich and Carlow, the concentration of inorganic C was determined by removing all organic carbon at a temperature of 450°C for 16 hours, followed by C analyses similar to total C by dry combustion in an elemental analyser (VarioMax, Hanau, Germany). Organic C concentrations were determined by difference between total and inorganic C. SOC contents in Carlow grassland were indirectly estimated by loss-of-ignition (LOI) at 500 ºC. The LOI values were multiplied by a conversion factor of 0.55

(Hoogsteen et al., 2015) to obtain the SOC concentration. SOC stocks were then calculated at each site by multiplying the SOC concentrations by the bulk density measured at each soil layer. Total stocks were obtained by summing up stocks of each layer to the maximum depth of measurement.

#### 2.3.2. Model parameterization

The soil carbon stocks in a LSM such as ORCHIDEE-SOM depend on the primary production (Todd-Brown et

al., 2013). Consequently, prior to the evaluation of the soil module in ORCHIDEE-SOM, we used gross primary production (GPP) measurements from the FLUXNET network to optimize the GPP-related parameters ($V_{cmax}$, Surface leaf area, maximum leaf area index, minimum leaf area index to start photosynthesis and minimum and maximum photosynthesis temperature sensitivity) in ORCHIDEE in order to assure that model inputs coming





from plant production are correct (Table S1). The ORCHIDEE data assimilation system, based on a Bayesian optimization scheme, was used for the optimization (MacBean et al., 2016). The optimization approach relies on the iterative minimization of the mismatch between the set of experimental observations and corresponding model outputs by adjusting the model driving parameters using L-BFGS-B algorithm (Byrd et al., 1995).

The simulations were done by using the default parameter set of ORCHIDEE-SOM (Table 1), which are defined based on prior knowledge and values reported in literature. Only the DOC decomposition times and microbial $CUE_{DOC}$ parameters were adjusted to site-specific conditions. For the Hainich forest and the Carlow grassland and cropland, where litter decomposes faster, the decomposition time of the stable DOC was assumed to be equal to the decomposition time of the labile DOC, that is, 1.3 days (see section 3.3.). Microbial $CUE_{DOC}$ is a PFT-

dependent parameter in the model that ranges from 0.3 to 0.55 (Table 1). While the mean measured CUE for soil microbial communities is 0.55, the recommendation for broad spatial scale models operating at long time steps is to use a CUE value of 0.3 (Manzoni et al., 2012; Sinsabaugh et al., 2013). Certainly, CUE shows a large variability and associated uncertainty, particularly in communities of terrestrial soils, as it is affected by multiple environmental factors and nutrient availability (Manzoni et al., 2012). Consequently, we selected the $CUE_{DOC}$

value within the reported range that performed best for each site simulation, namely 0.35 for Hainich and 0.5 for the other three sites.

### 2.3.3. Model simulations on site

The performance of the model was tested using data of DOC and SOC stocks at the four selected sites. Since these sites are all part of the FLUXNET network (Baldocchi et al., 2001), the in-situ measured meteorological variables

were available to be used as forcing for the simulations in ORCHIDEE. For Carlow grassland, only one year (2008) of flux measurements was available. However, Carlow grassland and cropland sites are located very close to each other and we therefore used the meteorological measurements of Carlow cropland as forcing for Carlow grassland. For all sites, the in-situ meteorological data was gap-filled using the ERA-interim 3-hourly product, following the methodology in Vuichard and Papale (2015).

Before the model application, we ran the model over approximately 14,000 years iteratively using the meteorological data for the available period for each site until all the soil variables reached a steady state (spin-up). The atmospheric $CO_2$ concentration was held at 350 ppm (Keeling and Whorf, 2006). For pH, clay content and bulk density, site-specific observed values were used (Table 2). The state of the ecosystem at the last time step of the spin-up was then used as the initial state for the simulations over the four selected sites for the period with

available flux measurements (Table 2). The site simulations were run at a daily time step to better explore the DOC temporal variations.

Goodness of fit for the monthly DOC measurements was assessed by calculating the coefficient of variation of the NRMSE (%), which is RMSE divided by the mean values of measurements, and then comparing the NRMSE values with the measurements uncertainty measured as standard deviation (SD) in %. The goodness of fit of the

model was defined as follows: "very good" for NRMSE < SD, "good" for SD<NRMSE<SD+30%, "fair" for SD+30%<NRMSE<SD+60%, and "bad" for NRMSE>SD+60% (Table S2).





### 3. Model results and discussion

#### 3.1. GPP

Accurate soil DOC simulations rely on the correct simulation of productivity, which in this study was approximated by optimizing the GPP modelled by ORCHIDEE-SOM at the study sites. After optimization of the
GPP-related parameters (Table S1), modelled and measured GPP were in good agreement for Brasschaat forest (modelled GPP of $1350 \pm 50$ g C m$^{-2}$ yr$^{-1}$ versus measured GPP of $1240 \pm 130$ g C m$^{-2}$ yr$^{-1}$), Hainich forest ($1410 \pm 70$ g C m$^{-2}$ yr$^{-1}$ versus $1520 \pm 110$ g C m$^{-2}$ yr$^{-1}$) and Carlow Cropland ($1250 \pm 180$ g C m$^{-2}$ yr$^{-1}$ versus $820 \pm 90$ g C m$^{-2}$ yr$^{-1}$) (Fig. 2). The GPP of Carlow grassland simulations could not be optimized due to limited data availability since GPP measurements were available only for the year 2008. Consequently, modelled GPP in 2008
was 40% higher than measured GPP for the single year of measurement, mainly due to a longer growing season modelled by ORCHIDEE-SOM. Maximum GPP values were similar between model and observations (Fig. 2C).

#### 3.2. SOC stocks and profiles

Overall, the simulated total SOC stocks to 2 m of depth were in good agreement with the measured values (Table 3). ORCHIDEE-SOM was able to simulate total SOC stocks within the standard deviation of the measurements at
the two forest sites (Fig. 3A and 3B), but overestimated SOC stocks by 6% in the grassland site and by 21% in the cropland site (Fig. 3C and 3D). Moreover, there was a good match between the measured and modelled SOC stocks at different soil depths at the four studied sites (Fig. 3), particularly at Brasschaat forest with measured depth to >1.5 m (Fig. 3A). The SOC stock vertical profile was thus very well reproduced by the model, suggesting that the two processes defining the SOC pools distribution in the model, i.e., 1) the vertical distribution of litter C
following the root C profiles, and 2) the vertical transport of SOC through diffusion, were properly represented in ORCHIDEE-SOM.

We assumed that soil fauna is present everywhere along the soil profile, and thus, the diffusion parameter was assumed to be constant along the soil profile in ORCHIDEE-SOM (Table 1). Because soil faunal activity may vary with depth, one could argue that the diffusion coefficient should be depth-dependent. However, models with
depth-dependent diffusion coefficients have been tested, but did not yield substantial improvements relative to models with a fixed diffusion coefficient (Boudreau, 1986). In fact, most of the models of diffusion at ecosystem level assume a constant diffusion parameter with depth (Bruun et al., 2007; Guimberteau et al., 2017; Obrien and Stout, 1978; Wynn et al., 2005). Our results confirm that this assumption is valid for four temperate sites with different vegetation inputs and soil properties.

#### 3.3. DOC dynamics at the site level

The ORCHIDEE-SOM simulation of DOC concentrations time series at the Brasschaat forest was in good agreement with the observed DOC concentrations in the intermediate soil layer (35 cm, Fig. 4B, NRMSE=35 %). However, it clearly overestimated the DOC concentrations in the upper soil layer (10 cm, Fig. 4A, NRMSE= 228%) and underestimated DOC concentrations in the subsoil (75 cm, Fig. 4C, NRMSE=58 %) (Table S2). This
coniferous forest showed the highest DOC concentrations (> 20 mg L$^{-1}$, Table 2), which partly originate from a "low-quality" litter, in this case needles, that decomposes more slowly and thus relatively more DOC remains in solution (Cotrufo et al., 2013; Zhang et al., 2008). The higher decomposition time for the stable DOC pool in the





model accounts for the slower decomposition of DOC in the Brasschaat forest, producing a good fit to the observations in the intermediate (35 cm) and subsoil layers (75 cm), but overestimating DOC concentrations in the upper soil layer. For the rest of the sites (the deciduous forest, the grassland and the cropland) where litter is more easily degradable, the decomposition time of the stable DOC was assumed to be equal to the labile DOC

pool, thus fitting the lower DOC concentrations at these sites.

In the case of Hainich forest, modelled DOC concentrations were in good agreement with the observed values, particularly in the upper soil layer (5 cm, Fig. 5A, NRMSE=30%). The model tended to underestimate DOC concentrations in deeper soil layers (10 and 20 cm, Fig. 5B and C, NRMSE=59 and 83%, respectively), but the modelled DOC concentrations were mostly still inside the standard deviations of the observations for all depths

(Fig. 5). In this case, the $CUE_{DOC}$ parameter was decreased from 0.5 to 0.35, value that is in agreement with observed bacterial growth efficiency in soil water matrix in a beech forest (Andreasson et al., 2009), to better capture the relatively low DOC concentrations observed in Hainich forest (median= 9.53 mg L$^{-1}$, range=1.5-50.7 mg L$^{-1}$). The higher soil pH compared to Brasschaat coniferous forest may have contributed to lowering DOC concentrations in the soil solution of the Hainich forest, and microbial CUE tends to decrease with soil pH,

reaching a minimum at pH 7.0 (Sinsabaugh et al., 2016).

Finally, DOC concentrations simulated in Carlow grassland were mostly in good agreement with measurements (Fig. 6, Table S2, NRMSE= 33-66 %), only at the beginning of 2007 the DOC concentrations in the topsoil were slightly overestimated (Fig. 6A). The model reproduced better the topsoil DOC concentrations measured in Box 2, while the best fit in the subsoil was for the DOC concentrations measured in Box 1 (Table S2). Moreover,

ORCHIDEE-SOM was able to reproduce well the DOC magnitude time series in the cropland site if we compare the simulated DOC at the 18-37 cm soil layer with the measurements from suction cups installed between 30 and 40 cm (Fig. 7, NRMSE= 29 %).

Even though the modelled DOC concentrations were overall within the range of variation of the measurements, ORCHIDEE-SOM was not able to fully capture the temporal dynamics of DOC concentrations at each site and

soil layer (Figs. 4-7). This is not surprising taking into account that soil DOC is the result of multiple interconnected environmental, biological and physico-chemical factors, and as a consequence, DOC variability is very high in time and space (Clark et al., 2010). For instance, measured DOC in Carlow grassland was slightly higher in the sampling position "Box 2" than in "Box1" (Fig. 6). Although these two sampling positions were only 150 meters apart, the small-scale soil heterogeneity and the gentle slope leading from Box1 to Box2 caused the

soil water contents to substantially differ between the two boxes (Fig. S4), leading to differences in DOC concentrations that cannot be captured in the model. Since a great proportion of DOC variability is explained by factors that are not accounted for in land surface models, such as metal complexion (Camino-Serrano et al., 2014), it is expected that daily DOC concentrations modelled by ORCHIDEE-SOM do not closely match spot measurements of DOC in soil solution. Nevertheless, at the four sites the magnitude of DOC concentrations was

overall well captured by ORCHIDEE-SOM, with the exception of the top layer in Brasschaat (Fig. 8), confirming the model applicability across a range of vegetation and soil types, after parameter adjustment for $CUE_{DOC}$ at Hainich and decomposition times of stable DOC in Hainich forest and Carlow grassland and cropland.

While the SOC profiles were well captured by ORCHIDEE-SOM, it was difficult to accurately simulate DOC concentrations along the different soil depths at site level using the default parameters. Interestingly, the model

was not very sensitive to parameters that directly affect the vertical distribution of carbon, such as the SOC and





DOC diffusion coefficient. On the contrary, it showed the greatest sensitivity to the biological parameters: the decomposition time of DOC and the $CUE_{DOC}$ (Table S3, Fig. S6 and S7). DOC in soils is primarily the result of enzymatic decomposition of litter and SOC, and it also originates from root exudates and from microbial residues which are not explicitly modelled. Simultaneously, microbial consumption of DOC is the main process of DOC

removal from soil (Bolan et al., 2011). Since both are biological processes, it explains the high model sensitivity to these biological parameters. DOC production and consumption are both controlled by the same factors that control biological activity, particularly temperature and moisture, and these processes will therefore vary with soil depth, land use type, and soil fertility (Bolan et al., 2011). However, these interactions are not represented in the model.

**3.4. Model limitations and further work**

In this study, ORCHIDEE-SOM was tested at four temperate ecosystems in Europe, but additional considerations need to be taken into account when applying the model to other ecosystems. For instance, the DOC coming from throughfall is not represented in ORCHIDEE-SOM, although it is a substantial source of soil DOC in tropical ecosystems (Lauerwald et al., 2017). Furthermore, the use of the hydrological module ORC11, that implies free

drainage in the bottom layer, currently limits the representation of more humid ecosystems such as wetlands or peatlands, where shallow water tables are present, and needs to be addressed in the near future.
Moreover, it is important to note that the modelled DOC along the soil profile strongly depends on the downward water flux simulated by the hydrological module ORC11. Therefore, the accuracy of the simulated DOC fluxes relies on the accuracy of the simulated soil water flux. However, while SWC is frequently measured in the field

(Figs. S2-S5), there are no site-level measurements of internal soil water fluxes, and therefore soil water fluxes between layers cannot be validated against observations, introducing a source of uncertainty in ORCHIDEE-SOM.
A general simplification of the model is that only the mineral soil is explicitly represented. The aboveground litter is dimensionless and the products of its decomposition are redistributed among the first five soil layers. As a consequence, these layers contain the organic material, but do not explicitly account for the substantial different

hydrological, chemical and physical processes that occur between organic materials and mineral soil layers. Although the current version of ORCHIDEE-SOM performed well at the Hainich forest (Fig. 5), the misrepresentation of the organic horizons (e.g., differentiation of humus types) may partly explain the deviance in DOC predictions in the upper soil layer of the Brasschaat forest (Fig. 4A). Overall, representing only the mineral soil layers may limit the model application in forests soil with potential large organic horizons, such as Podzols

and Gleysols or, more importantly, in organic soils such as peatlands.
Future model applications will require further empirical parameterization. Several studies highlight the importance of soil properties and vegetation characteristics in SOC-related parameters, such as the effect of soil type and litter decomposability (humus type) on microbes (Sulman et al., 2014) or the effect of soil texture, SOC content and bulk density on the moisture-soil respiration relationship (Moyano et al., 2012). We applied an empirical

relationship that links the adsorption coefficient ($K_D$) with soil properties (clay and pH) (Eq. 15), as a first step towards linking the physico-chemical soil properties to our model parameters, although the correlation was weak and some important parameters (Al or Fe in soils) are still missing. A similar exercise should be done for the biological model parameters like $CUE_{DOC}$, which for the moment do not reflect their known changes with vegetation or soil properties (Manzoni et al., 2017; Sinsabaugh et al., 2016). Also, the SOC diffusion coefficient





was kept constant along the soil profile, although it is known that diffusion is higher in the upper soil layers, and that biotic activity is controlled by the pH, among other factors (Jagercikova et al., 2014). Our findings thus point out to the necessity of a depth- and vegetation dependent parameterization of the new model parameters in the future.

Finally, our results showed that the model performed worst at Brasschaat, which is the site with the most extreme soil conditions (very acidic and sandy soil). It is not surprising that ORCHIDEE-SOM, which is designed for regional or global simulations, is unable to well reproduce SOC and DOC dynamics at sites with particular characteristics because it uses many default parameters based on prior knowledge (Table 1).

Additional parameter optimization through data assimilation at multisite level is thus needed before applying
ORCHIDEE-SOM to large scale simulations. Calibration of the new parameters of ORCHIDEE-SOM by assimilation of DOC concentration and SOC stocks data from several sites across different biomes will not only make the model applicable to large scale simulations, but also will give insight to the relative importance of processes affecting SOC and DOC in different ecosystem types (Braakhekke et al., 2013) and will reduce the uncertainty range of the new parameters, which is an essential part of any process-based large scale model (Zaehle
et al., 2005).

We present here a new soil module within the land surface model ORCHIDEE. This module keeps the pool-based structure of the CENTURY model, but is upgraded to represent the biological production and consumption, mineral sorption and transport of vertically discretized SOC and DOC. None of the existing soil models that represent all these DOC-related processes and that vertically discretized SOC are embedded within a land surface
model. ORCHIDEE-SOM is an intermediate complexity model that, albeit its simplicity and generalization, has proven successful in simulating soil solution DOC concentrations at four different ecosystem types (Fig. 8). Once the soil module in ORCHIDEE-SOM is optimized for large-scale simulations and linked to the already existing DOC river scheme (Lauerwald et al., 2017), it will improve the predictions of the SOC vulnerability to climate change and the predictions of the present and future contribution of the aquatic continuum fluxes to the global C
cycle, thus improving the allocation of terrestrial and ocean C sinks.

## 4. Conclusions

ORCHIDEE-SOM is a new vertically resolved soil module, embedded in the land surface model ORCHIDEE, that represents litter, SOC and DOC dynamics and transport in and out of the soil. Key model improvements compared to the trunk version of ORCHIDEE are that ORCHIDEE-SOM can simulate deep SOC dynamics
(vertical profiles of SOC) and loss of organic carbon through leaching. We evaluated the model for four European sites with different vegetation covers, using input parameters that are realistic compared to prior knowledge. The modelled SOC stock profiles agree very well with the observations. Overall, the model was able to reproduce DOC concentrations at different soil depths, although DOC concentrations were overestimated in the upper horizon at the coniferous forest.

Moving forward requires an exhaustive model parameterization. Our results suggest that empirical data should be integrated in the SOC and DOC decomposition times and $CUE_{DOC}$ parameters in order to make them soil- and vegetation- dependent. Moreover, to be able to run ORCHIDEE-SOM at regional or global scales, the new parameters should be optimized by data assimilation and the optimized model needs testing against observations of SOC and DOC at larger scales (continental or global).





With our work on ORCHIDEE-SOM, we prepared the necessary model structure to simulate deep SOC and DOC dynamics by implementing the processes of DOC production and decomposition, DOC adsorption/desorption in mineral soils, SOC bioturbation and DOC transport with water flux. Although the current model still requires exhaustive parameterization, we conclude that by improving each of these new model elements we have the opportunity to end up with a robust, albeit simple and general, global tool for prediction of soil carbon fluxes and leaching to rivers and lakes.

### Code availability

The SVN version of the code branch is https://forge.ipsl.jussieu.fr/orchidee/browser/branches/ORCHIDEE-SOM revision 4407 and is available upon request. The ORCHIDEE data assimilation tool is available through a dedicated web site: https://orchidas.lsce.ipsl.fr.

### Author contributions

MC, B. Guenet, IJ, SL, PC and JP conceived the study. MC and B. Guenet performed all the simulations. RL and A. J-P optimized the code and assisted with technical aspects of the model. VB provided and assisted with the optimization tool (ORCHIDAS) and NV provided and assisted with the preparation of the forcing files using FLUXNET data. BdV, B. Gielen, GG, MS and DW provided with the site measurements and other site information needed for the model evaluation. KK and DK provided with the adsorption isotherms datasets and SV assisted with the statistics for the adsorption function. All the authors wrote the paper.

### Competing interests

The authors declare that they have no conflict of interest.

*Acknowledgements.* This work used eddy covariance data acquired and shared by the FLUXNET community, including these networks: AmeriFlux, AfriFlux, AsiaFlux, CarboAfrica, CarboEuropeIP, CarboItaly, CarboMont, ChinaFlux, Fluxnet-Canada, GreenGrass, ICOS, KoFlux, LBA, NECC, OzFlux-TERN, TCOS-Siberia, and USCCC. The ERA-Interim reanalysis data are provided by ECMWF and processed by LSCE. The FLUXNET eddy covariance data processing and harmonization was carried out by the European Fluxes Database Cluster, AmeriFlux Management Project, and Fluxdata project of FLUXNET, with the support of CDIAC and ICOS Ecosystem Thematic Center, and the OzFlux, ChinaFlux and AsiaFlux offices. MCS, PC, JP and IJ acknowledge funding from the European Research Council Synergy grant ERC-2013-SyG-610028 IMBALANCE-P. RL acknowledges funding from the European Union's Horizon 2020 research and innovation program under grant agreement no.703813 for the Marie Sklodowska-Curie European Individual Fellowship "C-Leak". BGuenet acknowledges funding from the ANR-14-CE01-0004 DeDyCAS project. SV is a postdoctoral research associate of the Fund for Scientific Research – Flanders.





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




Tables

Table 1 - List of parameters of ORCHIDEE-SOM (the name used in the model is between brackets), with their description, value, units, the parameterization used for each parameter.

| Parameter | Description | Value | Units | Parameterization |
|---|---|---|---|---|
| **Fixed** | | | | |
| Z_litter (*z_litter*) | Thickness litter above | 10 | mm | Assumed |
| **Soil carbon parameters** | | | | |
| frac_carb_ap (*frac_carb_ap*) | fraction of the active pool going into the passive pool | 0.004 | - | Parameterization based on Parton et al., 1987 |
| frac_carb_sa (*frac_carb_sa*) | fraction of the slow pool going into the active pool | 0.93 | - | Parameterization based on Parton et al., 1987 |
| frac_carb_sp (*frac_carb_sp*) | fraction of the slow pool going into the passive pool | 0.03 | - | Parameterization based on Parton et al., 1987 |
| frac_carb_pa (*frac_carb_pa*) | fraction of the passive pool going into the active pool | 1 | - | Parameterization based on Parton et al., 1987 |
| frac_carb_ps (*frac_carb_ps*) | fraction of the passive pool going into the slow pool | 0 | - | Parameterization based on Parton et al., 1987 |
| active_to_pass_clay_frac (*active_to_pass_clay_frac*) | | 0.68 | | Parton et al., 1987 |
| carbon_tau active (*carbon_tau_iactive*) | Decomposition times in carbon pools | 1 | years | This study |
| carbon_tau slow (*carbon_tau_islow*) | Decomposition times in carbon pools | 6.0 | years | This study |
| carbon_tau passive (*carbon_tau_ipassive*) | Decomposition times in carbon pools | 462.0 | years | This study |
| priming_param (c) active (*priming_param_iactive*) | Priming parameter for mineralization active | 493.66 | | (Guenet et al., 2016) |
| priming_param (c) slow (*priming_param_islow*) | Priming parameter for mineralization slow | 194.03 | | (Guenet et al., 2016) |
| priming_param (c) passive (*priming_param_ipassive*) | Priming parameter for mineralization passive | 136.54 | | (Guenet et al., 2016) |
| FLUX_TOT_COEFF (*flux_tot_coeff*) | Coefficient modifying the fluxes (1.2 and 1.4 increase decomposition due to tillage, 0.75 modify the flux depending on clay content) | 1.2, 1.4, .75 | days | (Gervois et al., 2008) for 1.2 and 1.4; Parton et al. (1987) for 0.75 |
| D (*Dif*) | Diffusion coefficient used for bioturbation litter and soil carbon | 2.74E-7 | $m^2$ $day^{-1}$ | (Koven et al., 2013) |
| **DOC parameters** | | | | |
| DOC_tau_labile (*DOC_TAU_LABILE*) | Decomposition time of labile DOC | 1.3 | days | Value within the range found in literature for fast pool of DOC (Boddy et al., 2008; Kalbitz et al., 2003; Qualls and Haines, 1992; Turgeon, 2008) |
| DOC_tau_stable (*DOC_TAU_STABLE*) | Decomposition time of stable DOC | 60.4 1.3* | days | Value within the range found in literature (Boddy et al., 2007; Boddy et al., 2008; Kalbitz et al., 2003; Qualls |

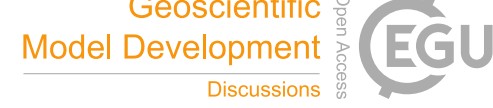

| | | | | and Haines, 1992; Turgeon, 2008) |
|---|---|---|---|---|
| D_DOC ($D\_DOC$) | Diffusion coefficient used for DOC diffusion | 1.0627E-5 | $m^2$ $day^{-1}$ | (Burdige et al., 1999) in (Ota et al., 2013) |
| $CUE_{DOC}$ ($CUE$) | Percentage of DOC decomposed that releases to $CO_2$ | PFT-dependent value: range 0.3-0.55 | - | Manzoni et al. (2012) (Sinsabaugh et al., 2013) |
| $K_D$ ($kd\_ads$) | Distribution coefficient of adsorbed DOC | Statistical relationship (Eq. 15) | $m^3$ water $kg^{-1}$ soil | Statistical relationship based on Kaiser et al., (1996) data |

\* The default *DOC_tau_stable* value in the model is 60.4 days, but *DOC_tau_stable* was adjusted to 1.3 days for the runs in the Hainich forest, the Carlow grassland and cropland.



Table 2. Characteristics of the four sites used for the model validation.

| **Site characteristics** | | | | |
|---|---|---|---|---|
| **Site** | Brasschaat | Hainich | Carlow | Carlow |
| **Ecosystem** | Coniferous forest *Pinus sylvestris* | Broadleaved forest *Fagus sylvatica* | Grassland Perennial ryegrass and white clover. | Cropland *Hordeum vulgare* L. cv. Tavern |
| **PFT** | 4: Temperate needleleaf evergreen trees | 6: Temperate broadleaf summergreen trees | 10: Natural C3 grass | 12: Agricultural C3 grass |
| **Soil properties*** | | | | |
| **Soil classification** | Arenosol | Cambisol | Luvisol | Cambisol |
| **pH** | 4 | 6.7 | 7.3 | 7.6 |
| **Clay** | 3.4 | 58.9 | 15 | 23 |
| **BD** | 1.4 | 1.2 | 1.2 | 1.55 |
| **DOC measurements** | | | | |
| **Measurements depths (cm)** | 10, 35, 75 | 5, 10 20 | Topsoil (10-30), Subsoil (60-75) | 40 |
| **Mean [DOC] per soil depth** | 39.2, 30.4, 22.3 | 16.7, 9.2, 6.6 | 8.5, 3.6 | 4.2 |
| **# Suction cups per horizon** | 6, 6, 6 | 4, 4, 4 | 10, 10 | 10 |
| **Measurement period** | 2000-2008 | 2001-2014 | 2006-2009 | 2006-2009 |
| **References** | (Gielen et al., 2011) | (Kindler et al., 2011) | (Kindler et al., 2011) | (Kindler et al., 2011; Walmsley et al., 2011) |
| **Meteorological observations** | | | | |
| **Forcings** | FLUXNET (1997-2010) | FLUXNET (2000-2007) | FLUXNET (2004-2008) | FLUXNET (2004-2008) |

* Soil properties are averaged over the soil profile.





Table 3. Simulated and observed mean soil organic carbon (SOC) stocks in the selected sites. Simulated SOC stocks were calculated down to the sampling depth at each site. Values are means ±SD.

| Site | Sampling depth (cm) | Soil Carbon Stocks (kg C m$^{-2}$) | |
|---|---|---|---|
| | | Measured | Modelled |
| Brasschaat | 100 | 12.1±3.1 | 14.15 |
| Hainich | 60 | 12.4±1.54 | 12.95 |
| Carlow grassland | 40 | 16.6 | 17.6 |
| Carlow cropland | 60 | 9.41±1.5 | 11.4 |


Figures

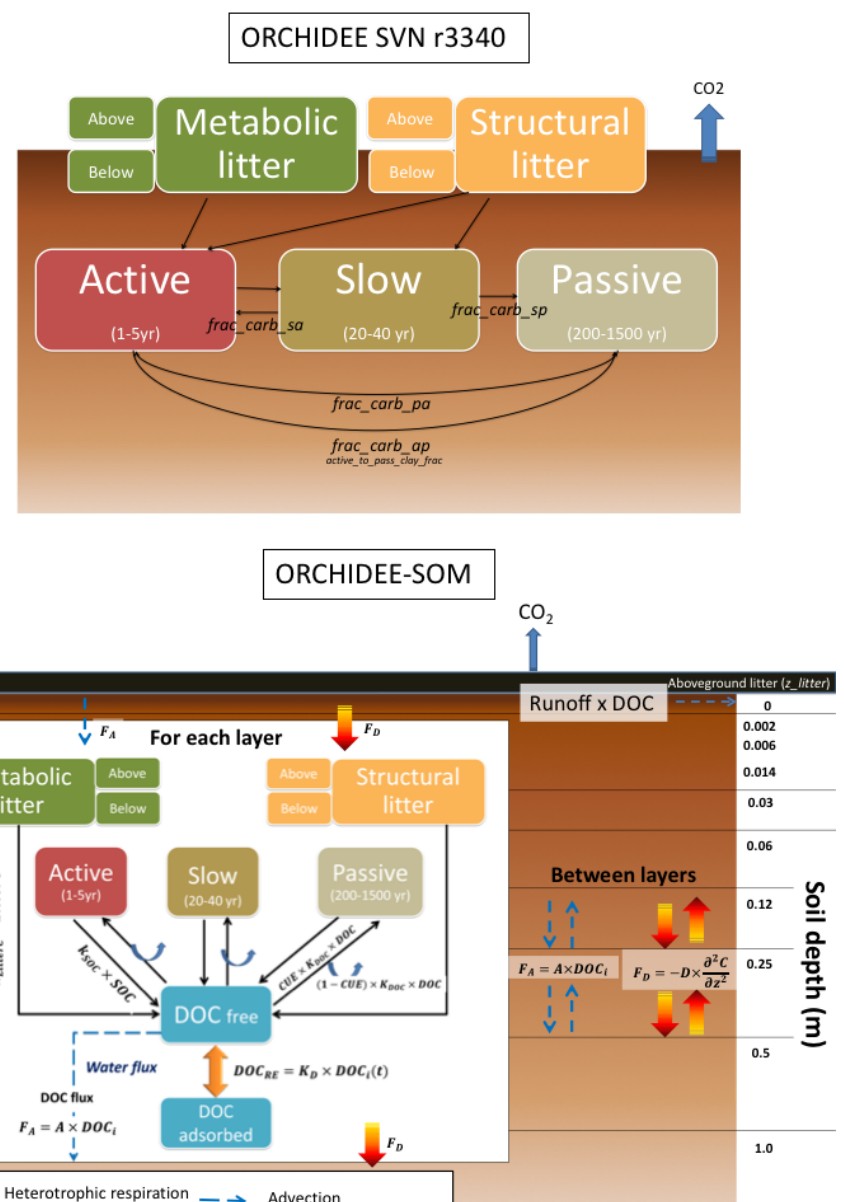

Figure 1.   Overview of the revised version of ORCHIDEE-SOM presented here (lower panel) compared to the soil module in the trunk version of ORCHIDEE SVN r3340 (upper panel). The white box represents pools, fluxes and major processes occurring in each of the 11 soil layers. The equations used for the processes occurring within and between layers are represented (see text for details).



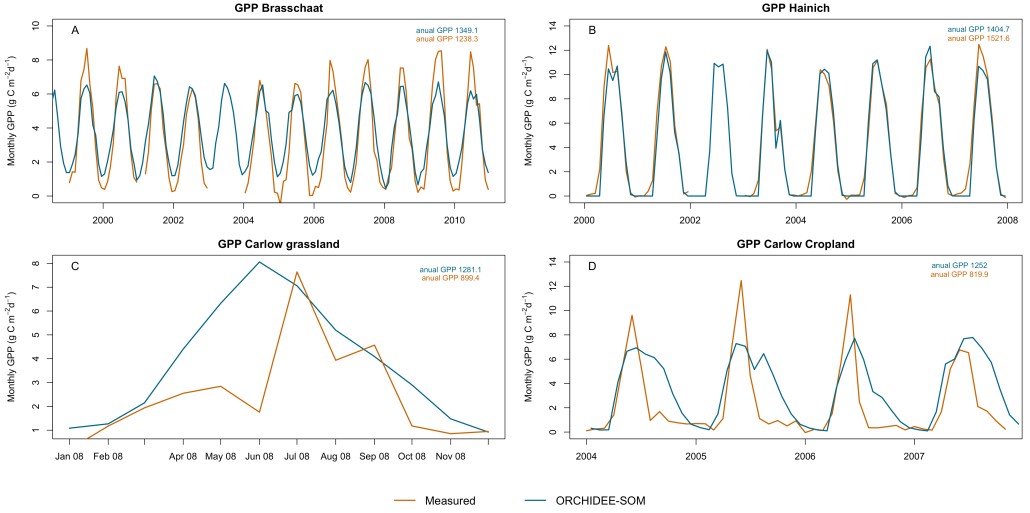

Figure 2. Measured and modelled daily GPP (monthly means) at the four sites, (A) Brasschaat forest, (B) Hainich forest, (C) Carlow grassland and (D) Carlow cropland, after GPP optimization. Measured GPP are acquired from each FLUXNET site and modelled GPP was acquired using ORCHIDEE-SOM.



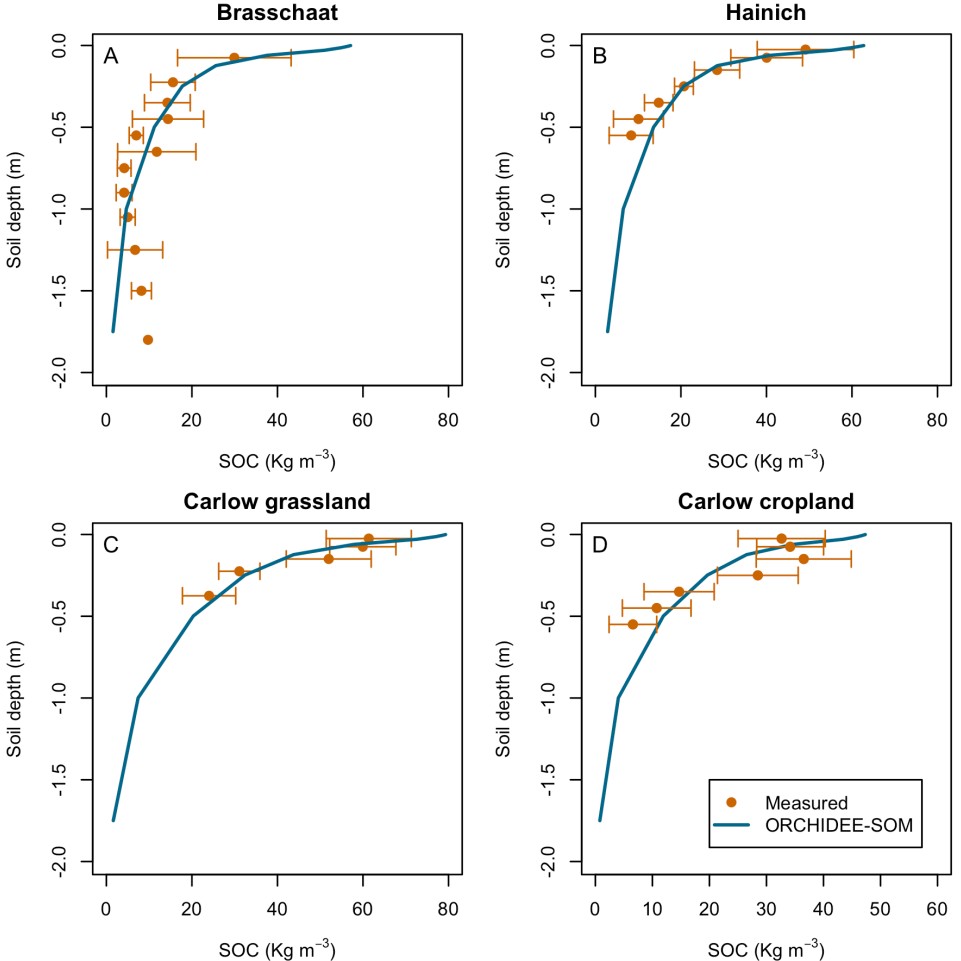

Figure 3. ORCHIDEE-SOM simulated (in blue) and measured (in brown) soil organic carbon profiles at the four sites, (A) Brasschaat forest, (B) Hainich forest, (C) Carlow grassland and (D) Carlow cropland. Error bars represent the standard deviation of the measurements.





Figure 4. ORCHIDEE-SOM simulated (daily values) and measured DOC dynamics at Brasschaat forest at three
soil depths, (A) 10 cm, (B) 35 cm and (C) 75 cm. Error bars are not displayed due to data unavailability.





Figure 5. ORCHIDEE-SOM simulated (daily values) and measured DOC dynamics at Hainich forest at three soil depths, (a) 5 cm, (b) 10 cm and (c) 20 cm. Error bars represent the standard deviation of site measurements (n=4).



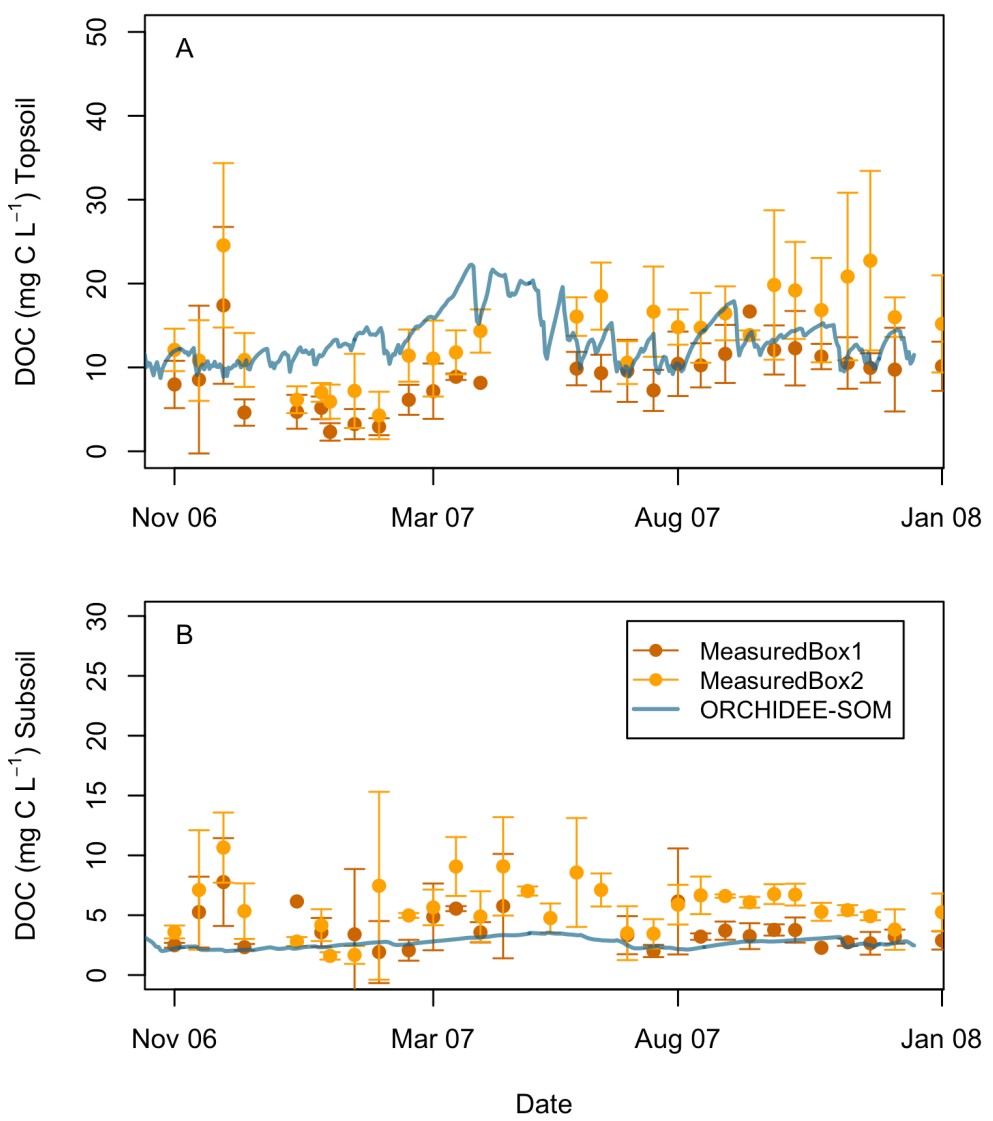

Figure 6. ORCHIDEE-SOM simulated (daily values) and measured DOC dynamics at Carlow grassland at two soil layers, (a) topsoil, 10-30 cm; and (b) subsoil, 60-75 cm. Site measurements are in brown for Box 1 and in orange for Box 2, and simulation results using ORCHIDEE-SOM (daily values) in blue. Error bars represent the standard deviation of site measurements (n=5).





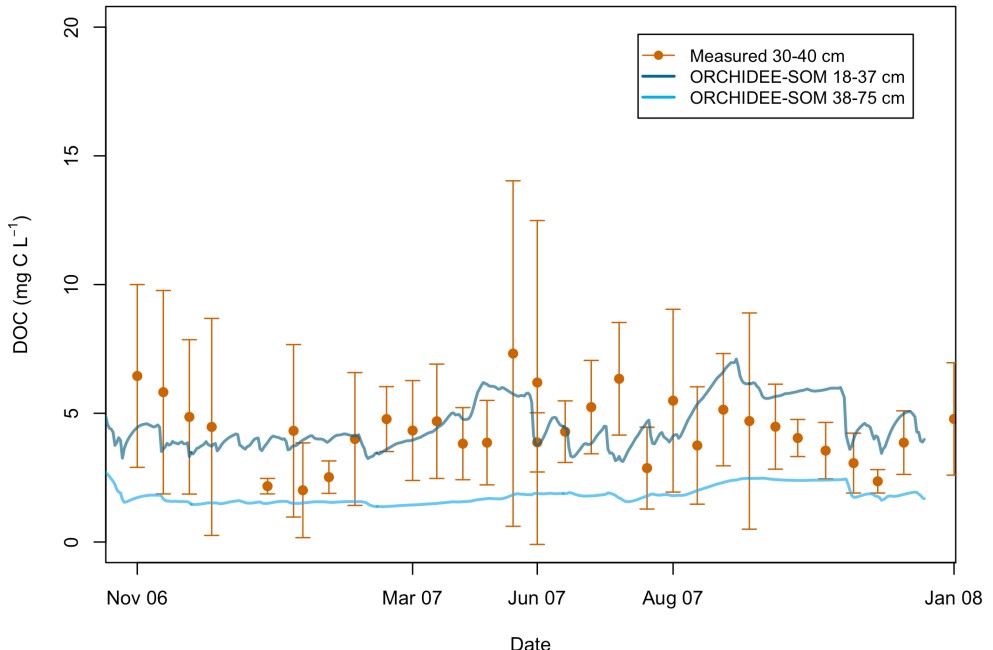

Figure 7. ORCHIDEE-SOM simulated (daily values) and measured DOC dynamics at Carlow cropland measured between 30 and 40 cm. Error bars represent the standard deviation of site measurements (n=10).





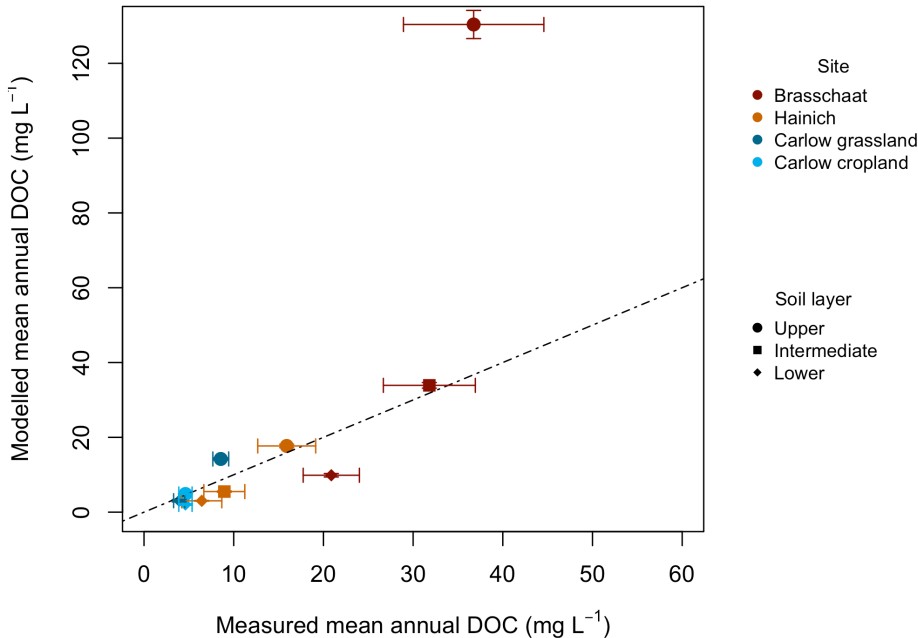

Figure 8. Measured versus modelled mean annual DOC concentrations (mg C L$^{-1}$). Error bars represent the standard deviations for the mean annual DOC concentrations. The dashed line represents the 1:1 line.