# Peer review of "ORCHIDEE-SOM: Modeling soil organic carbon (SOC) and dissolved organic carbon (DOC) dynamics along vertical soil profiles in Europe"

_Geoscientific Model Development, 2017_

## Referee Comment (RC1) · Anonymous Referee #1 · 20 Dec 2017

Camino-Serrano et al. developed a soil module in ORCHIDEE to represent the vertically resolved SOC and DOC based on the pool-based structure of the CENTURY model. It is encouraging to see such kind of model development as noted by the authors that explicit modeling of DOC within and export from the land soil is challenging, but is important for the accurate estimation of the global SOC stocks. One of my concerns is the assumption of "all the products of decomposition from litter and SOC go to free DOC". I hope the authors can provide some support for this assumption, either from prior knowledge or from model optimization for simulated DOC and SOC. My other comments are listed below.

[Figure]

1. Change "decomposition times" to "turnover times" to be consistent with common terminology used in land models.

2. The definition of k in the equations should be decomposition rate constants, not rates.

3. Eqs. (2), (5), and (7) use the same I(t) term for input of each pools. They need to be differentiated and explicitly written out for readers to reproduce the model. For example, in Eq.(7), I(t) = ksol x SOC + klit x litter.

4. Suggest adding the gama term defined in Eq. (6) into Eq. (5).

5. The units for DOC are not consistent when defining the biological processes (e.g. Eq. (5)) and sorption (Eq. 12). State it if conversion is needed.

6. P7, line 17 – Is the "sum" within each soil layer?

7. P9, line 31 - What "boundary conditions"?

8. P11, lines 18,19 – Remove "in" in the parenthesis.

9. P13, line 25 – Change "iteratively" to "repeatedly".

10. P14 – Please explain the early decrease of GPP from the measurement compared to the simulation in Figure 2D.

11. P15, line 13,14 – From Eq. 15, the clay content rather than pH can better explain the lower DOC. Was the statement about CUE here used to explain the lower DOC too? If so, from Eq. (7), decrease of CUE actually increases DOC.

12. P16, line 19 – Define SWC.

13. Any attempt made to optimize the simulated DOC and test if the proposed model makes sense?
* * *
GMDD

Interactive
comment

---

## Referee Comment (RC2) · Anonymous Referee #2 · 7 Jan 2018

This is a well written manuscript documenting the improvements made in modeling soil carbon dynamics in the ORCHIDEE land surface model. Authors modeled the vertical SOC and DOC dynamics, and suggest for data assimilation in order to optimize the model parameters for regional and global application of ORCHIDEE- SOM. The findings of this manuscript are directly relevant to the readership of the GMD journal. I have provided my comments/suggestions from the perspective of a soil scientist.

- At many places in the manuscript, authors mentioned that this study also modeled the lateral transport of SOC and DOC from soil to rivers and lakes. But while looking at proposed equations, I could not figure out how was that done? To appropriately model

the lateral transport of SOC, DOC (which is an important component of SOC dynamics) authors need to use surface hydrology equations that uses the role of topographic features in lateral transport of carbon and other nutrients. In general, this model could be valuable if authors can vary the mentioned soil processes vary along with the soil forming factors (eg. Topography, vegetation types, climate, soil wetness etc.). The data of environmental data of soil forming factors are available globally at a variety of resolutions.

- Soil depth determines the volume of SOC and DOC at certain location. I don't think an assumption of homogenous soil depth of 2m globally is valid. Because the soil depth may range from few centimeters to many meters depending upon the certain location and its soil type.

- P5 L10-15: Can these layers represent soil horizons? Soil horizons determine the physical, chemical, biological properties of soils at a certain depth. It will be ideal if the depth discretization can at least try to mimic the soil horizons.

- P6L10: In my knowledge none of the global models represent pedogenic processes that are important to soil carbon dynamics such as podzolization, organo-mineral complexation, clay migration, soil aggregation, pedoturbation etc. I don't expect authors to represent all of these processes in this paper, but it will be reasonable if authors can indicate in their limitations section that these processes are important to soil C dynamics and are not represented in land surface models currently.

- P6L17: What about surface organic layer thickness, which is present in forest, grassland, and tundra ecosystems?

- P6L25-26: I don't think its a reasonable assumption to think that the above ground litter layer is a fixed parameter. It should vary at least according to the land cover types or Pfts.

- P6L30: How this depth relates with the actual soil depth?

[Figure]

- P7L1: I think the control of soil moisture and temperature on SOC decomposition should vary along with other soil-forming factors.

―――――――――――――――――――

---

## Author Comment (AC1) · 26 Jan 2018

Response to Reviewers

Note: All the references to pages and lines numbers in this response correspond to the revised version of the manuscript attached to this letter.

Reviewer #1:

Camino-Serrano et al. developed a soil module in ORCHIDEE to represent the vertically resolved SOC and DOC based on the pool-based structure of the CENTURY model. It is encouraging to see such kind of model development as noted by the au-

thors that explicit modeling of DOC within and export from the land soil is challenging, but is important for the accurate estimation of the global SOC stocks. One of my concerns is the assumption of "all the products of decomposition from litter and SOC go to free DOC". I hope the authors can provide some support for this assumption, either from prior knowledge or from model optimization for simulated DOC and SOC. My other comments are listed below.

Thank you very much for your comments.

The model assumption that we adopted, where "all the products of decomposition from litter and SOC go to free DOC" is based on the hypothesis that the soluble state is a prerequisite for the degradation of soil organic matter. This assumption has been stated previously (Marschner and Kalbitz, 2003). Water availability is necessary for the release of nutrients from soil organic matter. In fact, all the processes of mineralization of organic matter are hydrolytic. According to this theory, dissolved organic carbon is the substrate for microorganisms and thus the products of decomposition experience a "soluble" phase.

More in detail, in the first stage of decomposition, water-soluble compounds leach out of the litter. By definition, any water-soluble compound leaching out will be free DOC. In the following stages of litter decomposition, extra-cellular enzymes depolymerize the structural molecules, producing water soluble monomers and/or oligomers, thus these products can also be considered as free DOC.

Therefore, we defined our modelling framework by taking into account the already existing pools of SOC in ORCHIDEE (based on the CENTURY model) and the theory here presented. To our minds, adding a DOC pool that will be the intermediate pool for products of decomposition was a feasible, albeit reasonable, assumption. We have added a sentence justifying this assumption in the revised manuscript (P7L34-36).

1. Change "decomposition times" to "turnover times" to be consistent with common terminology used in land models.

"Decomposition times/rates" have been changed to "turnover times/rates" in the revised manuscript.

2. The definition of k in the equations should be decomposition rate constants, not rates.

It has been corrected.

3. Eqs. (2), (5), and (7) use the same I(t) term for input of each pools. They need to be differentiated and explicitly written out for readers to reproduce the model. For example, in Eq.(7), I(t) = ksol x SOC + klit x litter.

The I(t) terms have been now differentiated in equations (2), (5) and (7) for clarity.

4. Suggest adding the gamma term defined in Eq. (6) into Eq. (5).

The gamma term defined in Eq. (6) applies only for the active pool, while Eq. (5) generally represents the SOC decomposition for all the SOC pools (active, slow and passive). That is why we presented the gamma term in a separate equation.

5. The units for DOC are not consistent when defining the biological processes (e.g. Eq. (5)) and sorption (Eq. 12). State it if conversion is needed.

The reviewer is right. The equations of sorption were not updated according to the latest version of the code. These have been corrected in the revised manuscript (P9L21-27) and now the units for DOC are consistent.

6. P7, line 17 – Is the "sum" within each soil layer?

Yes, LOC is the stock of labile organic C defined as the sum of the C pools with a higher turnover rate than the pool considered within each soil layer. It has been clarified in the manuscript (P7L22-24).

7. P9, line 31 - What "boundary conditions"?

By "boundary conditions" we meant "external factors" to be used for the soil module

of ORCHIDEE. We have changed the sentence to "... these variables are not globally available and hence not included in the land surface model ORCHIDEE" to avoid confusion (P9L38-39).

8. P11, lines 18,19 – Remove "in" in the parenthesis.

This has been corrected.

9. P13, line 25 – Change "iteratively" to "repeatedly".

This has been corrected.

10. P14 – Please explain the early decrease of GPP from the measurement compared to the simulation in Figure 2D.

The early decrease of GPP from the measurements compared to the modelled GPP of the cropland site (Figure 2D) may be explained by the harvesting time of the spring barley established at this site. The C3 crop PFT in the default version of ORCHIDEE is assumed to have the same phenology as natural grasslands, but with higher carboxylation rates (Krinner et al., 2005; Wu et al., 2016). Harvest is represented as a fixed ratio of aboveground biomass removed from the field at each time step. Consequently, the model is not capable to reproduce observed harvest. Therefore, the standard version of ORCHIDEE is not able to capture the early end of the growing season (decrease in GPP) caused by the crop harvesting, which in the case of spring barley occurs in summer.

11. P15, line 13,14 – From Eq. 15, the clay content rather than pH can better explain the lower DOC. Was the statement about CUE here used to explain the lower DOC too? If so, from Eq. (7), decrease of CUE actually increases DOC.

As the reviewer points out, Eq. 15 indicates a positive relationship between clay and the coefficient of adsorption, however a preliminary sensitivity analysis (data not shown) showed a low model sensitivity to the adsorption coefficient. In fact, the percentage of adsorbed DOC is always small compared to the free DOC, and consequently Eq. 15 is

not determinant in the differences in DOC magnitudes across sites. In the mentioned paragraph, we propose that the higher pH in Hainich may partly explain the lower DOC concentrations of the measurements, based on previous findings (e.g., Löfgren and Zetterberg, 2011) that suggest that larger DOC concentrations are found in more acidic soils.

We noted a mistake in the statement about the CUE in P15: "and microbial CUE tends to decrease with soil pH, reaching a minimum at pH 7.0 (Sinsabaugh et al., 2016)." should be ""and microbial CUE tends to decrease with increasing soil pH, reaching a minimum at pH 7.0 (Sinsabaugh et al., 2016)." It has been corrected in the manuscript. This statement was used to support our selection of a lower CUE parameter for the Hainich simulation (CUE=0.35), where soil pH is higher and close to 7.

We agree with the reviewer that this sentence was not clear for the reader. We have reformulated it in the revised manuscript (P15L16-19).

12. P16, line 19 – Define SWC.

SWC has been defined here.

13. Any attempt made to optimize the simulated DOC and test if the proposed model makes sense?

Yes, we tried the same Bayesian-based data assimilation exercise used for GPP optimization (the ORCHIDEE data assimilation system) but for the DOC concentrations at the three soil depths for Brasschaat, Hainich and Carlow grassland presented in this study. However, as explained in the manuscript, this optimization approach relies on the iterative minimization of the mismatch between the set of experimental observations and corresponding model outputs and thus it requires to perform thousands of simulations. For each simulation of soil DOC, we need to previously reach the equilibrium of the soil carbon pools by simulating thousands of years to make the model outputs comparable with data. This process is very time-consuming compared to the

simulations needed to optimize the aboveground carbon pools and fluxes, e.g., GPP, because aboveground C pools reach equilibrium after few years. Therefore, it was not possible to optimize the DOC and SOC with the available tools.

Nevertheless, we believe that the satisfactory performance of ORCHIDEE-SOM reproducing SOC stocks (Figure 3) and DOC concentrations (Figure 8) supports the validity of our proposed model structure. Therefore, yes, although an exhaustive parameterization is still needed to validate the model for larger-scale applications, these supports show that the proposed model "makes sense".

Reviewer #2:

This is a well written manuscript documenting the improvements made in modeling soil carbon dynamics in the ORCHIDEE land surface model. Authors modeled the vertical SOC and DOC dynamics, and suggest for data assimilation in order to optimize the model parameters for regional and global application of ORCHIDEE- SOM. The findings of this manuscript are directly relevant to the readership of the GMD journal. I have provided my comments/suggestions from the perspective of a soil scientist.

Thanks for the positive assessment.

1) At many places in the manuscript, authors mentioned that this study also modeled the lateral transport of SOC and DOC from soil to rivers and lakes. But while looking at proposed equations, I could not figure out how was that done? To appropriately model the lateral transport of SOC, DOC (which is an important component of SOC dynamics) authors need to use surface hydrology equations that uses the role of topographic features in lateral transport of carbon and other nutrients. In general, this model could be valuable if authors can vary the mentioned soil processes along with the soil forming factors (e.g. Topography, vegetation types, climate, soil wetness etc.). The data of environmental data of soil forming factors are available globally at a variety of resolutions.

In ORCHIDE-SOM, the lateral transport of SOC and DOC from soil to rivers is simply calculated by multiplying the sum of the DOC concentrations in the first 5 soil layers by the runoff flux and the DOC concentration in the bottom layer by the drainage flux, as described in section 2.2.3. (P12, L5-7). The equation for DOC exported out of the soil, not included in the manuscript will be:

$DOCexp = DOC(z=1-5)*R+DOC(z=1)*F+DOC(z=11)*D$

where R is the runoff, F is the flooding and D is the drainage flux, and DOC is the free DOC concentration at soil layer z.

In ORCHIDEE, the runoff is produced by infiltration excess. The infiltration rate depends on precipitation rates, local slope and vegetation and is limited by the hydraulic conductivity of the soil. Therefore, ORCHIDEE calculates a Hortonian surface runoff (d'Orgeval et al., 2008; Lauerwald et al., 2017). This sentence has been added to the manuscript's paragraph describing the calculation of runoff and drainage in ORCHIDEE (P11, L29-32).

2) Soil depth determines the volume of SOC and DOC at certain location. I don't think an assumption of homogenous soil depth of 2m globally is valid. Because the soil depth may range from few centimeters to many meters depending upon the certain location and its soil type.

We fully agree with the reviewer. The actual soil depth is determinant for the simulation of SOC and DOC stocks. However, ORCHIDEE-SOM is a land surface model designed for global simulations and currently there is no information on actual soil depths globally. Hence, soil modules within land surface models usually fix a maximum soil depth in their model framework (e.g., Koven et al., 2013), as we do in ORCHIDEE-SOM.

Nevertheless, even if we had information on the actual soil depth for each site, the soil hydrological module in ORCHIDEE-SOM calculates the water balances and fluxes based on a 2-m depth soil column and thus the soil hydrology is highly impacted by

the soil depth parameter. Then, we should be cautious when changing the maximum soil depth within the model. Moreover, since the SOC and DOC dynamics are partially controlled by the hydrological module, we consider that parameterizing the C related parameter using a different soil depth than the one used for large scale simulation may induce trouble for our next step, which is performing global simulations.

This concern has been addressed by adding a final paragraph in section 2.1. in the revised manuscript (P6, L20-24).

3) P5 L10-15: Can these layers represent soil horizons? Soil horizons determine the physical, chemical, biological properties of soils at a certain depth. It will be ideal if the depth discretization can at least try to mimic the soil horizons.

This is a good point. We are aware that, from an edaphic point of view, representing soil horizons is more accurate than simply dividing the soil profile into fixed soil layers according to the depth, because the physico-chemical soil properties at a certain soil depth will vary largely among sites. However, as explained later in the manuscript (P6, L1-6), we adopted the vertical discretization used for the soil hydrology scheme for technical reasons. The geometrical configuration used in the hydrological module is commonly used in most land surface models for describing the vertical soil water fluxes. In ORCHIDEE-SOM, these vertical soil water fluxes are needed for the calculation of DOC concentrations and fluxes and, in order to match the soil C and water fluxes, we adopted the same vertical discretization. Trying to group the soil hydrology layers to represent soil horizons will imply making important assumptions and generalizing the depth intervals of each soil horizon, but it is known that soil horizon's thicknesses will vary among sites/soil profiles. In conclusion, we decided that the best option was to adopt the hydrological discretization. The soil layers in ORCHIDEE-SOM can still be matched to soil horizons by soil depth for each site-specific case. Nevertheless, a sentence about the misrepresentation of actual soil horizons has been added in the discussion (P16, L35-38).
4) P6L10: In my knowledge, none of the global models represent pedogenic processes that are important to soil carbon dynamics such as podzolization, organo-mineral complexation, clay migration, soil aggregation, pedoturbation etc. I don't expect authors to represent all of these processes in this paper, but it will be reasonable if authors can indicate in their limitations section that these processes are important to soil C dynamics and are not represented in land surface models currently.

A paragraph with the idea that important pedogenic processes are still missing in the land surface models has been added in the discussion section "Model limitations and further work" (P17, L3-9).

5) P6L17: What about surface organic layer thickness, which is present in forest, grassland, and tundra ecosystems?

By aboveground litter layer, we mean the surface organic/litter layers in all the different PFTs. Therefore, the sentence "In ORCHIDEE-SOM, a new parameter to define the thickness of the aboveground litter layer (z_litter), assumed constant over time, has been added to allow the calculation of aboveground litter diffusion into the mineral soil (Table 1)." applies also to the surface organic layer thickness of forest, grassland and tundra ecosystems.

6) P6L25-26: I don't think it's a reasonable assumption to think that the above ground litter layer is a fixed parameter. It should vary at least according to the land cover types or Pfts.

We agree with the reviewer that the thickness of the litter layer is very important when modelling DOC. However, there are two reasons for fixing this parameter:

1) ORCHIDEE-SOM is intended for global simulations and, as it occurs with the soil depth (see answer to comment 2 above), there is no information on the thickness of the litter layers globally.

2) Actually, the parameter for the thickness of litter layer was included just to allow the

calculation of the diffusion, but ORCHIDEE-SOM does not explicitly represent the litter layer: the aboveground litter layer is dimensionless, which means that processes of production and decomposition of aboveground litter occur independently of the litter layer thickness in the model (see section 2.1). Moreover, this parameter is subject to a great temporal variation, particularly in PFTs such as broadleaved forests (Hainich). In order to actually take into account the effect of the thickness of the litter layer, a 3D model for the litter that will relate the thickness of the litter layer with the phenology will be needed, but this is missing in ORCHIDEE-SOM at the moment.

Furthermore, a preliminary sensitivity analysis showed a low model sensitivity to the z_litter parameter: increasing the z_litter parameter by a factor of 20 produced just a moderate change in simulated free DOC concentrations (30% change) (data not shown). For all these reasons, we keep z_litter constant.

7) P6L30: How this depth relates with the actual soil depth?

For the moment, this depth is the maximum soil depth in the model, fixed to 2 meters in ORCHIDEE-SOM (see answer to comment 2).

8) P7L1: I think the control of soil moisture and temperature on SOC decomposition should vary along with other soil-forming factors.

First of all, the control of soil moisture and temperature on SOC decomposition is not completely independent on the soil factors, because the soil moisture and soil temperature depends on the clay content.

However, we are aware of the importance of the sensitivity of SOC decomposition to soil moisture and temperature and its interactions with soil properties. For instance, it is known that temperature sensitivity of SOC decomposition varies with mean residence times (Davidson and Janssens, 2006) and there is an on-going work on changing the Q10 value in ORCHIDEE-SOM based on the turnover rate.

Regarding the soil moisture, we also worked on implementing the empirical relationship developed by Moyano et al., (2012) that relates the response of heterotrophic respiration to soil moisture with other soil properties, namely, soil texture, organic carbon content and bulk density. However, the use of this relationship lead to complex feedbacks between SOC and soil moisture that are unresolved for the moment. In conclusion, ORCHIDEE-SOM represents an improvement compared to other land surface models due to new processes represented, such as the adsorption of DOC on soil minerals, or the vertical movement of SOC and DOC, but we agree that there is plenty of room for model improvement regarding important soil and environmental relationships that has been proved empirically but that are still not included in our model. This point is discussed in the section "Model limitations and further work" (P17, L10-22).

References

Camino-Serrano, M., Gielen, B., Luyssaert, S., Ciais, P., Vicca, S., Guenet, B., De Vos, B., Cools, N., Ahrens, B., Arain, M. A., Borken, W., Clarke, N., Clarkson, B., Cummins, T., Don, A., Pannatier, E. G., Laudon, H., Moore, T., Nieminen, T. M., Nilsson, M. B., Peichl, M., Schwendenmann, L., Siemens, J. and Janssens, I.: Linking variability in soil solution dissolved organic carbon to climate, soil type, and vegetation type, Global Biogeochem. Cycles, 28(5), 497–509, doi:Doi 10.1002/2013gb004726, 2014.

d'Orgeval, T., Polcher, J. and de Rosnay, P.: Sensitivity of the West African hydrological cycle in ORCHIDEE to infiltration processes, Hydrol. Earth Syst. Sci., 12(6), 1387–1401, 2008.

Davidson, E. A. and Janssens, I. A.: Temperature sensitivity of soil carbon decomposition and feedbacks to climate change, Nature, 440(7081), 165–173, doi:Doi 10.1038/Nature04514, 2006.

Koven, C. D., Riley, W. J., Subin, Z. M., Tang, J. Y., Torn, M. S., Collins, W. D., Bonan, G. B., Lawrence, D. M. and Swenson, S. C.: The effect of vertically resolved soil biogeochemistry and alternate soil C and N models on C dynamics of CLM4, Biogeosciences, 10(11), 7109–7131, doi:DOI 10.5194/bg-10-7109-2013, 2013.

Krinner, G., Viovy, N., de Noblet-Ducoudre, N., Ogee, J., Polcher, J., Friedlingstein, P., Ciais, P., Sitch, S. and Prentice, I. C.: A dynamic global vegetation model for studies of the coupled atmosphere-biosphere system, Global Biogeochem. Cycles, 19(1), doi:Artn Gb1015Doi 10.1029/2003gb002199, 2005.

Lauerwald, R., Regnier, P., Camino-Serrano, M., Guenet, B., Guimberteau, M., Ducharne, A., Polcher, J. and Ciais, P.: ORCHILEAK: A new model branch to simulate carbon transfers along the terrestrial-aquatic continuum of the Amazon basin, Geosci. Model Dev. Discuss., 1–58, doi:10.5194/gmd-2017-79, 2017.

Löfgren, S. and Zetterberg, T.: Decreased DOC concentrations in soil water in forested areas in southern Sweden during 1987-2008, Sci. Total Environ., 409(10), 1916–1926, doi:DOI 10.1016/j.scitotenv.2011.02.017, 2011.

Marschner, B. and Kalbitz, K.: Controls of bioavailability and biodegradability of dissolved organic matter in soils, Geoderma, 113(3–4), 211–235, doi:10.1016/S0016-7061(02)00362-2, 2003.

Moyano, F. E., Vasilyeva, N., Bouckaert, L., Cook, F., Craine, J., Yuste, J. C., Don, A., Epron, D., Formanek, P., Franzluebbers, A., Ilstedt, U., Katterer, T., Orchard, V., Reichstein, M., Rey, A., Ruamps, L., Subke, J. A., Thomsen, I. K. and Chenu, C.: The moisture response of soil heterotrophic respiration: interaction with soil properties, Biogeosciences, 9(3), 1173–1182, doi:DOI 10.5194/bg-9-1173-2012, 2012.

Sinsabaugh, R. L., Turner, B. L., Talbot, J. M., Waring, B. G., Powers, J. S., Kuske, C. R., Moorhead, D. L. and Follstad Shah, J. J.: Stoichiometry of microbial carbon use efficiency in soils, Ecol. Monogr., 86(2), 172–189, doi:10.1890/15-2110.1, 2016.

Wu, X., Vuichard, N., Ciais, P., Viovy, N., de Noblet-Ducoudré, N., Wang, X., Magliulo, V., Wattenbach, M., Vitale, L., Di Tommasi, P., Moors, E. J., Jans, W., Elbers, J., Ceschia, E., Tallec, T., Bernhofer, C., Grünwald, T., Moureaux, C., Manise, T., Ligne, A., Cellier, P., Loubet, B., Larmanou, E. and Ripoche, D.: ORCHIDEE-CROP (v0), a

new process-based agro-land surface model: model description and evaluation over Europe, Geosci. Model Dev., 9(2), 857–873, doi:10.5194/gmd-9-857-2016, 2016.

Please also note the supplement to this comment:
https://www.geosci-model-dev-discuss.net/gmd-2017-255/gmd-2017-255-AC1-supplement.pdf